# Towards Understanding Why Label Smoothing Degrades Selective Classification and How to Fix It

**Guoxuan Xia,**[1,†] **Olivier Laurent,**[2,3,4] **Gianni Franchi**[2] **& Christos-Savvas Bouganis**[1]
Imperial College London,[1] U2IS, ENSTA, Institut Polytechnique de Paris,[2]
SATIE, Université Paris-Saclay,[3] DTIS, ONERA[4]

## Abstract

Label smoothing (LS) is a popular regularisation method for training neural networks as it is effective in improving test accuracy and is simple to implement. "Hard" one-hot labels are "smoothed" by uniformly distributing probability mass to other classes, reducing overfitting. Prior work has suggested that in some cases *LS can degrade selective classification (SC)* – where the aim is to reject misclassifications using a model's uncertainty. In this work, we first demonstrate empirically across an extended range of large-scale tasks and architectures that LS *consistently* degrades SC. We then address a gap in existing knowledge, providing an *explanation* for this behaviour by analysing logit-level gradients: LS degrades the uncertainty rank ordering of correct vs incorrect predictions by suppressing the max logit *more* when a prediction is likely to be correct, and *less* when it is likely to be wrong. This elucidates previously reported experimental results where strong classifiers underperform in SC. We then demonstrate the empirical effectiveness of post-hoc *logit normalisation* for recovering lost SC performance caused by LS. Furthermore, linking back to our gradient analysis, we again provide an explanation for why such normalisation is effective. Project page: [https://ensta-u2is-ai.github.io/Understanding-Label-smoothing-Selective-classification/](https://ensta-u2is-ai.github.io/Understanding-Label-smoothing-Selective-classification/)

## 1 Introduction

Label smoothing (LS) (Szegedy et al., 2016) is a common regularisation technique used to improve classification accuracy in deep learning. The one-hot labels used for cross entropy (CE) are linearly combined with a uniform distribution over classes, redistributing the probability mass and "smoothing" the "hard" targets, discouraging overfitting. Due to its simplicity and empirical effectiveness, label smoothing features in many recent training recipes (Vaswani et al., 2017; Tan et al., 2019; He et al., 2019; Touvron et al., 2021; Liu et al., 2021; 2022b;c; Tan & Le, 2019; 2021), being particularly popular on the ImageNet-1k (Russakovsky et al., 2015) image classification benchmark.

Within the domain of uncertainty estimation, LS is well explored in the context of model calibration (Müller et al., 2019; Chun et al., 2020; Mukhoti et al., 2020; Liu et al., 2022a), where the aim is to align a model's output probabilities with its empirical accuracy. The pairing is intuitive as LS encourages models to output lower probabilities, and models are typically more confident than they are accurate (Guo et al., 2017). On the other hand, there is very little research investigating the effects of LS in the context of *Selective Classification*. Selective Classification (SC) (Hendrycks & Gimpel, 2017; Geifman & El-Yaniv, 2017; Xia & Bouganis, 2022b; Jaeger et al., 2023) is a problem setting where, in addition to the primary classification task, a binary rejection decision is made based on the uncertainty estimated by the model, *i.e. reject/abstain if uncertain*. The aim is to reduce the number of failures served by the classifier by pre-emptively rejecting potential misclassifications. It is well motivated by applications where safety and reliability are important due to the high cost of failure. For example, when uncertain, an autonomous driving system may require driver assistance (Kendall & Gal, 2017), a medical diagnosis system may defer to a doctor (Beam & Kompa, 2021; Kurz et al., 2022), or a visual aid system for the visually impaired may abstain from answering a query (Whitehead et al., 2022).

Recently Zhu et al. (2022; 2024) empirically observe that LS can lead to worse SC for convolutional neural networks (CNNs) performing image classification, *yet it is not clear why this occurs*. Concurrently, large-scale empirical evaluations of open-source pre-trained models have shown

---

[†]corresponding author – g.xia21@imperial.ac.uk

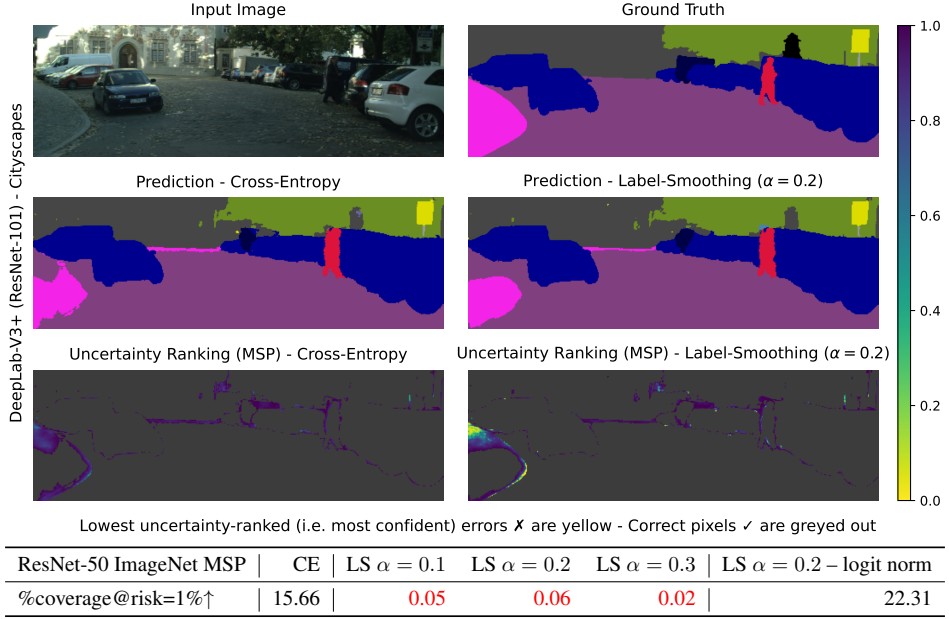

| ResNet-50 ImageNet MSP | CE | LS $\alpha = 0.1$ | LS $\alpha = 0.2$ | LS $\alpha = 0.3$ | LS $\alpha = 0.2$ – logit norm |
|---|---|---|---|---|---|
| %coverage@risk=1%↑ | 15.66 | 0.05 | 0.06 | 0.02 | 22.31 |

Figure 1: **Top**: LS causes overconfidence for semantic segmentation. The LS-trained model predicts much lower (ranked) uncertainty on incorrect ✗ segmentations than CE. In particular, **for the erroneous region on the left where the model has predicted parts of the "sidewalk" as "road", the LS model is highly overconfident**. This could have dire consequences in a safety-critical application such as autonomous driving. **Bottom**: LS leads to close to 0% of samples being accepted (coverage) when a strict tolerance of 1% error on accepted samples (risk) is imposed on ImageNet. Deployment-time logit normalisation effectively negates the degradation caused by LS.

that many strong classifiers have surprisingly poor SC ability (Galil et al., 2023; Cattelan & Silva, 2024). Cattelan & Silva (2024) additionally present deployment-time logit normalisation as a sometimes-effective approach to improving SC, *but the reason for its effectiveness remains unclear*. In this work, we aim to empirically validate and analytically demystify the effect of LS on SC, tying together and filling in the gaps in knowledge of previous work with the following **key contributions**:

1. We show empirically, across a range of large-scale architectures (CNN,ViT) and tasks (image classification, semantic segmentation), that *training with LS consistently leads to degraded SC performance* (see Fig. 1), even if it may improve accuracy. Moreover, we find that the degradation worsens with stronger LS. As LS can be found in the training recipes of many of the models evaluated in (Galil et al., 2023; Cattelan & Silva, 2024), this suggests LS as one potential cause for previously unexplained negative results where strong classifiers underperform at SC.

2. We address a gap in the understanding of LS by providing an explanation of this behaviour through analysing the logit-level gradients of the LS loss. We show that the amount LS suppresses the max logit directly corresponds to the true probability of error $P_{\text{error}}$, with suppression *increasing(decreasing)* the more likely a prediction is correct(wrong). This leads to relatively *higher* uncertainty on correct predictions and *lower* uncertainty on misclassifications, degrading the ranking of uncertainties and hurting SC compared to vanilla CE.

3. We show that post-hoc *logit normalisation* (Cattelan & Silva, 2024) is effective in negating the degradation from LS (Fig. 1). Moreover, we elucidate this effectiveness through the lens of our gradient-based analysis. Linking back to the imbalanced logit suppression of LS, we find that logit normalisation compensates for this effect by increasing uncertainty as the max logit increases.

## 2 PRELIMINARIES

For a glossary of notation see Appendix A. Consider a $K$-class neural network classifier with parameters $\boldsymbol{\theta}$ that models the conditional distribution $P(y|\boldsymbol{x};\boldsymbol{\theta})$ of labels $y \in \mathcal{Y} = \{\omega_k\}_{k=1}^K$ given inputs $\boldsymbol{x} \in \mathcal{X} = \mathbb{R}^D$. Typically the network has a categorical softmax output $\boldsymbol{\pi}(\boldsymbol{x};\boldsymbol{\theta}) \in [0,1]^K$,

$$P(\omega_k|\boldsymbol{x};\boldsymbol{\theta}) = \pi_k(\boldsymbol{x};\boldsymbol{\theta}) = \exp v_k(\boldsymbol{x}) / \sum_{i=1}^K \exp v_i(\boldsymbol{x}) , \quad \boldsymbol{v} = \boldsymbol{W}\boldsymbol{z} + \boldsymbol{b} , \tag{1}$$

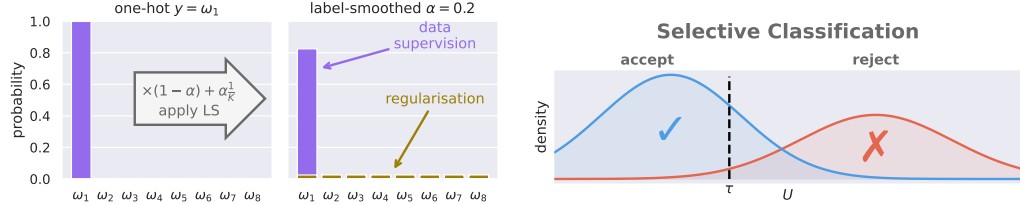

Figure 2: **Left**: illustration of how **label smoothing** (LS) alters a training label. LS reduces data supervision and adds regularisation, potentially improving generalisation by reducing overfitting. **Right**: illustration of **selective classification** (SC). Uncertain samples ($U > \tau$) are rejected/detected, to reduce the number of errors ✗ served by the system. Rejected samples can be discarded or processed separately (*e.g.* deferred to a human expert). We wish to better separate/rank ✓ vs ✗ via $U$.

where $\boldsymbol{v} \in \mathbb{R}^K$ are the logits output by the final layer with weight matrix $\boldsymbol{W} \in \mathbb{R}^{K \times L}$, bias $\boldsymbol{b} \in \mathbb{R}^K$, and pre-logit features $\boldsymbol{z} \in \mathbb{R}^L$ as inputs. The neural network is trained by minimising the cross entropy (CE) loss on a finite dataset $\mathcal{D}_{\mathrm{tr}} = \{x^{(n)}, y^{(n)}\}_{n=1}^N$ drawn from distribution $p_{\mathrm{data}}(\boldsymbol{x}, y)$, such that it approximately learns the true conditional $P_{\mathrm{data}}(y|\boldsymbol{x})$,

$$\mathcal{L}_{\mathrm{CE}}(\boldsymbol{\theta}) = -\frac{1}{N} \sum_n \sum_k \delta_{y^{(n)} \omega_k} \log P(\omega_k | \boldsymbol{x}^{(n)}; \boldsymbol{\theta}) \tag{2}$$

$$\approx -\mathbb{E}_{p_{\mathrm{data}}(\boldsymbol{x})} \left[ \sum_k P_{\mathrm{data}}(\omega_k | \boldsymbol{x}) \log P(\omega_k | \boldsymbol{x}; \boldsymbol{\theta}) \right] \tag{3}$$

$$= \mathbb{E}_{p_{\mathrm{data}}(\boldsymbol{x})} \left[ \mathrm{KL} \left[ \bar{\boldsymbol{\pi}}(\boldsymbol{x}) || \boldsymbol{\pi}(\boldsymbol{x}; \boldsymbol{\theta}) \right] \right] + \mathrm{const.} = \mathcal{L}_{\mathrm{CE}}^{\mathrm{true}}(\boldsymbol{\theta}) , \tag{4}$$

where $\delta_{ij} = 1$ if $i = j$, and $0$ if $i \neq j$ is the Kronecker delta and $\mathrm{KL}[\cdot||\cdot]$ is the Kullback–Leibler divergence. We use $\bar{\boldsymbol{\pi}}(\boldsymbol{x}) \in [0,1]^K$ as a shorthand for the true conditional categorical, *i.e.* $\bar{\pi}_k = P_{\mathrm{data}}(\omega_k | \boldsymbol{x})$. Predictions $\hat{y}$ are then made on new unlabelled input data $\boldsymbol{x}^*$ using classifier function $f$ during deployment,

$$\hat{y} = f(\boldsymbol{x}^*; \boldsymbol{\theta}) = \arg \max_\omega P(\omega | \boldsymbol{x}^*; \boldsymbol{\theta}) . \tag{5}$$

We also define the probability of the classifier making an error on a given sample as $P_{\mathrm{error}} = 1 - \bar{\pi}_{\hat{y}}$ , where in a slight abuse of notation $\bar{\pi}_{\hat{y}}$ is the true probability of the predicted class $P_{\mathrm{data}}(\hat{y}|\boldsymbol{x})$.

## 2.1 LABEL SMOOTHING (LS)

Label smoothing involves mixing the original one-hot labels ($\delta$ in Eq. (2)) with a uniform categorical distribution $\boldsymbol{u} = 1/K \cdot \mathbf{1}$ using hyperparameter $\alpha \in [0, 1]$ (see Fig. 2). The LS loss is thus

$$\mathcal{L}_{\mathrm{LS}}(\boldsymbol{\theta}; \alpha) = -\frac{1}{N} \sum_n \sum_k \left[ (1 - \alpha) \delta_{y^{(n)} \omega_k} + \alpha \frac{1}{K} \right] \log P(\omega_k | \boldsymbol{x}^{(n)}; \boldsymbol{\theta}) \tag{6}$$

$$= (1 - \alpha) \mathcal{L}_{\mathrm{CE}}(\boldsymbol{\theta}) + \alpha \frac{1}{N} \sum_n \left[ \mathrm{KL} \left[ \boldsymbol{u} || \boldsymbol{\pi}(\boldsymbol{x}^{(n)}; \boldsymbol{\theta}) \right] \right] + \mathrm{const.} \tag{7}$$

$$\approx \mathbb{E}_{p_{\mathrm{data}}(\boldsymbol{x})} \left[ \mathrm{KL} \left[ \underbrace{(1 - \alpha) \bar{\boldsymbol{\pi}}(\boldsymbol{x})}_{\text{data supervision}} + \underbrace{\alpha \boldsymbol{u}}_{\text{regularisation}} || \boldsymbol{\pi}(\boldsymbol{x}; \boldsymbol{\theta}) \right] \right] + \mathrm{const.} = \mathcal{L}_{\mathrm{LS}}^{\mathrm{true}}(\boldsymbol{\theta}; \alpha) , \tag{8}$$

where we see that it can also be viewed as *reduced* CE supervision from the data combined with a regularisation term encouraging the softmax $\boldsymbol{\pi}$ to be uniform and preventing it from overfitting to the training data (Fig. 2). Eq. (8) also shows that LS can be seen as learning to predict a "softened" version of the true conditional $(1 - \alpha) \bar{\boldsymbol{\pi}}(\boldsymbol{x}) + \alpha \boldsymbol{u}$, encouraging a model to be less confident on *all* samples.

## 2.2 SELECTIVE CLASSIFICATION (SC)

A simple downstream task for estimates of predictive uncertainty is to reject (or detect) predictions that may incur a high cost (Xia & Bouganis, 2022b; Jaeger et al., 2023), using a *binary rejection function*,

$$g(\boldsymbol{x}; \tau) = \begin{cases} 0 \ (\text{reject prediction}), & \text{if } U(\boldsymbol{x}) > \tau \ (\text{uncertain}) \\ 1 \ (\text{accept prediction}), & \text{if } U(\boldsymbol{x}) \leq \tau \ (\text{confident}) , \end{cases} \tag{9}$$

where $U(\boldsymbol{x})$ is a scalar measure of predictive uncertainty extracted from the prediction model and $\tau$ is a user-set operating threshold. Intuitively, we reject if the model is uncertain, preventing the system from serving failures, which may then be processed separately. We are only concerned with the *relative rankings* of $U$s rather than absolute values - we want to be *more uncertain* on failures. In the case where the prediction task is *classification* and we wish to reject potential misclassifications (✗), we can use a *selective classifier* (Chow, 1970; El-Yaniv & Wiener, 2010) $(f, g)$, which is simply the combination of a classifier $f$ (Eq. (5)) and the aforementioned binary rejection function $g$ (Eq. (9)). Fig. 2 contains an illustration. To evaluate a selective classifier, we use the 0/1 classification error,

$$\mathcal{L}_{\mathrm{SC}}(f(\boldsymbol{x}), y) = \begin{cases} 0, & \text{if } f(\boldsymbol{x}) = y \quad (\text{correct } ✓) \\ 1, & \text{if } f(\boldsymbol{x}) \neq y \quad (\text{misclassified } ✗) \,, \end{cases} \quad (10)$$

to define the *selective risk* (El-Yaniv & Wiener, 2010; Geifman & El-Yaniv, 2017) as

$$\mathrm{Risk}(f, g; \tau) = \frac{\mathbb{E}_{p_{\mathrm{data}}(\boldsymbol{x}, y)}[g(\boldsymbol{x}; \tau)\mathcal{L}_{\mathrm{SC}}(f(\boldsymbol{x}), y)]}{\mathbb{E}_{p_{\mathrm{data}}(\boldsymbol{x}, y)}[g(\boldsymbol{x}; \tau)]} \,, \quad (11)$$

which is the average error on the *accepted* samples. The denominator of Eq. (11) is the proportion of samples accepted, or the *coverage*, $\mathrm{Cov} = \mathbb{E}_{p_{\mathrm{data}}(\boldsymbol{x})}[g(\boldsymbol{x}; \tau)]$. Our objective is to minimise risk for a given coverage (lower %error on accepted samples) and/or maximise coverage for a given risk (accept more samples). Note this can be achieved both through improving $f$ (fewer errors) and through improving $g$ (better rejection). SC performance is evaluated via the Risk-Coverage (RC) curve (Geifman & El-Yaniv, 2017) (see Fig. 3 for examples). The area under the curve (AURC↓) provides an aggregate metric over $\tau$,[1] whilst the curve can also be inspected at specific operating points (Whitehead et al., 2022; Xia & Bouganis, 2023), *e.g.* coverage at 5% risk (Cov@5↑). For deployment, $\tau$ can be set using a held-out validation dataset by finding a suitable operating point on the RC curve according to an external requirement *e.g.* risk=1% if tolerance for failure is low.

$$U(\boldsymbol{x}) = -\mathrm{MSP}(\boldsymbol{x}) = -\pi_{\max} = -\max_k \pi_k(\boldsymbol{x}; \boldsymbol{\theta}) = -P(\hat{y}|\boldsymbol{x}; \boldsymbol{\theta}) \,, \quad (12)$$

*i.e.* the (negative of the) Maximum Softmax Probability (MSP) (Hendrycks & Gimpel, 2017) is the model's estimate of the probability of its prediction $\hat{y}$. We focus on this uncertainty measure $U$ for SC as it is a popular default choice and has been shown to consistently and reliably perform well in the literature (Xia & Bouganis, 2022b; 2023; Jaeger et al., 2023; Feng et al., 2023) compared to alternatives. MSP is a natural choice as $P(y|\boldsymbol{x}; \boldsymbol{\theta})$ models $P_{\mathrm{data}}(y|\boldsymbol{x})$, and $U = -\max_k P_{\mathrm{data}}(\omega_k|\boldsymbol{x})$ (or any scalar monotonic to it) provides the *optimal* risk if the true distribution $P_{\mathrm{data}}(y|\boldsymbol{x})$ is known (Chow, 1970). We omit evaluations on other softmax-based $U$ from the main paper as we find them to behave similarly to MSP. In line with previous work (Xia & Bouganis, 2022b; Jaeger et al., 2023; Zhu et al., 2024), we find that OOD detection (Yang et al., 2021) scores perform poorly at SC, and also omit them (see Appendix F.1 for discussion and additional results).

**Over/Underconfidence.** Since we are only concerned about the *relative ranking* of $U$, we loosely refer to *overconfidence* as when a model's estimate of uncertainty for a given sample is (relatively) low when $P_{\mathrm{error}}$ high, as we want a model to be uncertain when it is likely to be wrong. *Underconfidence* is then the inverse. Both will result in worse SC: overconfidence leads to errors being accepted, underconfidence to correct predictions being rejected. Intuitively, over/underconfidence may arise from poorness of fit, where $\boldsymbol{\pi}(\boldsymbol{x}; \boldsymbol{\theta})$ fails to accurately model the true conditional distribution $\bar{\boldsymbol{\pi}}(\boldsymbol{x})$ (Fig. 4).

This is *different* to the definition used in model calibration (Guo et al., 2017), which, notably, is concerned with marginal properties (averaged over the input distribution) rather than per-sample properties, as well as absolute probabilities rather than relative rankings. It is possible to be highly over/underconfident in calibration but optimal for SC and vice versa (Zhu et al., 2024), *e.g.* applying the monotonic transformation $U = -\max_k[P_{\mathrm{data}}(\omega_k|\boldsymbol{x})]^{1000}$. We note that model calibration is an independent task to SC (Jaeger et al., 2023), is well explored in the context of LS (Müller et al., 2019), and is not the focus of this work. LS may improve calibration by increasing the uncertainty of *both* correct ✓ *and* incorrect ✗ predictions, however, this will not necessarily help SC (Zhu et al., 2024).

## 3 THE EFFECT OF LABEL SMOOTHING ON SELECTIVE CLASSIFICATION

Zhu et al. (2022; 2024) observe empirically (as part of a broader investigation) that for a single value of $\alpha$ LS degrades SC for CNN-based image classification. In this section we aim to validate this

---

[1] We note that there are a number of alternative aggregate metrics to AURC (Geifman et al., 2019; Cattelan & Silva, 2024; Traub et al., 2024). We choose to omit them, as we focus on non-aggregated results in this work.

behaviour on a wider range of large-scale tasks and architectures. We find that **LS leads to consistent degradation in SC**, with stronger LS leading to greater degradation. We suggest that it may partially explain the findings of recent large-scale empirical evaluations of open-source pre-trained models (Galil et al., 2023; Cattelan & Silva, 2024) where strong classifiers surprisingly *underperform on SC*.

**Experimental details.** We investigate large-scale[2] image classification on ImageNet-1k (Russakovsky et al., 2015) and semantic segmentation (pixel-level classification) on Cityscapes (Cordts et al., 2016). For ImageNet, we randomly split the validation set into 10,000 validation and 40,000 evaluation images. For Cityscapes, we randomly split the original validation set into 100 validation and 400 evaluation images. To estimate the risk for the RC curves, we subsample 5000 labelled pixels per image at random. We evaluate on the same pixels between models. The only parameter varied between training runs is the level of LS $\alpha$, and to isolate the effects of LS, we train all models from scratch using simple recipes. We purposely avoid augmentations such as MixUp (Zhang et al., 2018) and CutMix (Yun et al., 2019) as these directly affect the training labels, which would interfere with our experiments. For ImageNet classification, we train ResNet-50 (He et al., 2016) and ViT-S-16 (Dosovitskiy et al., 2021) using only random resized cropping and flipping for data augmentation. To achieve decent accuracy for ViT training from scratch without advanced augmentations, we use sharpness-aware minimisation (SAM) (Foret et al., 2021; Chen et al., 2022). For semantic segmentation on Cityscapes, we train DeepLabV3+ (Chen et al., 2018) (ResNet-101 backbone) using only random cropping, flipping and colour jitter for augmentations. Extended training details for reproducibility can be found in the Appendix C. We release our code on GitHub.

## 3.1 Label smoothing degrades selective classification

To examine the effects of LS on SC, we plot RC curves in Fig. 3. We use $U = -\text{MSP}$ and vary only the LS level $\alpha$ between training runs. We see that **training with LS leads to a noticeable degradation for selective classification**, with higher $\alpha$ worsening the effect. We provide illustrative examples of LS overconfidence on ImageNet in Fig. 9. Although LS leads to slightly better risk at higher coverage, it quickly becomes worse than CE as coverage is reduced. The degradation is especially evident for the low-risk regime, which is relevant to *safety-critical* scenarios where tolerances for error are strict. For example, if the target is to achieve only 1% error/risk on ImageNet, then *all* of our LS models have close to zero coverage, rendering them effectively useless (Figs. 1 and 3). To further highlight the overconfidence caused by LS, we visualise the uncertainty ranking of incorrect pixels for the segmentation of a Cityscapes scene in Fig. 1. The LS-trained model is extremely confident for an erroneous region where it has predicted the "sidewalk" as "road". Whilst the CE model has made the same error, it is much more uncertain. This illustrates potential danger in an autonomous driving scenario if the vehicle is making decisions based on uncertainty estimates.

Upon inspection, many of the SC-underperforming models benchmarked in (Galil et al., 2023; Cattelan & Silva, 2024) are trained using LS, aligning with our results. The models are sourced from repositories such as `torchvision` (Paszke et al., 2019) and `timm` (Wightman, 2019), where the training recipes are optimised for top-1 accuracy. We note that for ImageNet, LS is such a common technique that it is often used *by default*, and not even mentioned in papers, *e.g.* (Tan & Le, 2019; 2021). Overall, our results highlight that **only optimising for accuracy may result in negative downstream consequences** and that practitioners of SC need to be aware of the effects of their training recipes. We include additional discussion of existing benchmarks and model checkpoints in Appendix F.6.

## 4 Towards explaining the negative effect of LS on SC

In the following section, we attempt to delve deeper into *why* our empirical results (and those in (Zhu et al., 2024)) occur, shedding light on a previously unexplained phenomenon. It might intuitively seem odd that LS degrades SC performance as the transform applied to the true targets in Eq. (8) $(1 - \alpha)\bar{\pi}(x) + \alpha u$ reduces the max probability for all samples but does not change the *relative ranking*. However, this only carries over to the model when it is knowledgeable about the data, so it is well fit $\pi(x; \theta) \approx (1 - \alpha)\bar{\pi}(x) + \alpha u$. For such samples, the model has *learnt to be more uncertain*. However, we cannot assume that the model is equally well fit on *all* regions of the data distribution.[3]

---

[2]We additionally provide small-scale CIFAR, tabular and text experiments in Appendices B.3 to B.5.

[3]For the sake of simplicity we avoid the term epistemic uncertainty (see Appendix F.2 for brief discussion).

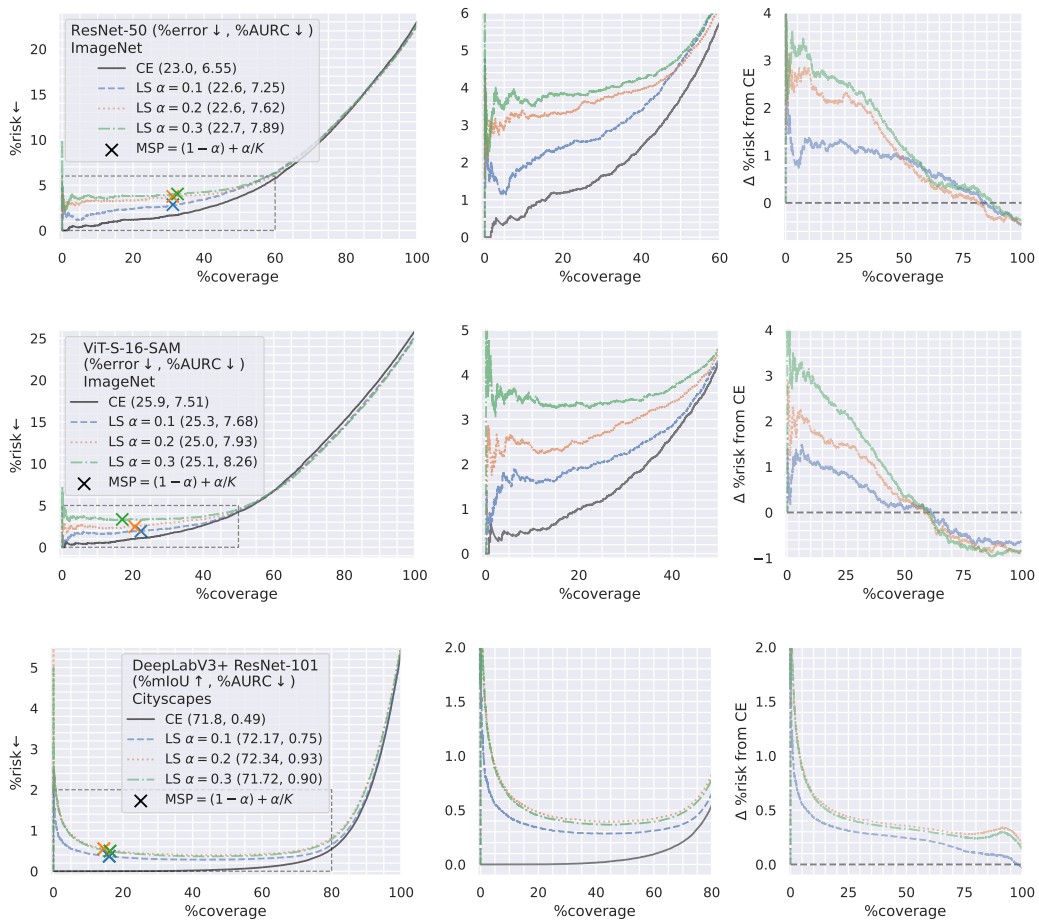

Figure 3: Risk-coverage plots for different levels of LS $\alpha$ for different models and tasks (ImageNet classification and Cityscapes semantic segmentation). Although it may improve error rate/accuracy at 100% coverage, **label smoothing consistently degrades SC performance.**

Through analysing the logit gradients, we provide a potential explanation: LS suppresses the max logit differently depending on how well fit the model is to a given sample – suppression is stronger(weaker) the more likely a model is correct(wrong), harming the model's ability to separate ✓ vs ✗.

## 4.1 COMPARING LOGIT GRADIENTS BETWEEN CE AND LS

To better understand how LS could lead to degraded SC, we consider how LS affects logit-level *training gradients*. These are the first term in the chain rule for backpropagation and so directly contribute to all parameter gradients during training. We take the gradient of $\mathcal{L}^{\text{true}}$ (Eqs. (4) and (8)),[4]

$$\frac{\partial \mathcal{L}^{\text{true}}_{\text{CE}}}{\partial v_k} = -\left[\bar{\pi}_k - \pi_k\right], \quad \frac{\partial \mathcal{L}^{\text{true}}_{\text{LS}}}{\partial v_k} = -\left[\left[\underbrace{(1-\alpha)\bar{\pi}_k}_{\text{data supervision}} + \underbrace{\alpha/K}_{\text{regularisation}}\right] - \pi_k\right], \quad (13)$$

for a single sample, where in a slight abuse of notation we omit the outer expectation over $p_{\text{data}}(\boldsymbol{x})$ for convenience. We can then define the **suppression gradient** on the logits,

$$\frac{\partial \mathcal{L}^{\text{true}}_{\text{sup}}}{\partial v_k} = \frac{\partial (\mathcal{L}^{\text{true}}_{\text{LS}} - \mathcal{L}^{\text{true}}_{\text{CE}})}{\partial v_k} = \frac{\partial \mathcal{L}^{\text{true}}_{\text{LS}}}{\partial v_k} - \frac{\partial \mathcal{L}^{\text{true}}_{\text{CE}}}{\partial v_k} = \alpha\left[\bar{\pi}_k - 1/K\right] = \alpha\bar{\pi}_k - \alpha/K, \quad (14)$$

---

[4]Here we assume that the empirical loss $\mathcal{L}$ (Eqs. (2) and (6)) approximates $\mathcal{L}^{\text{true}}$, in order to relate our discussion to $P_{\text{error}}$. However, the same analysis on the empirical loss leads to similar conclusions (see Appendix D).

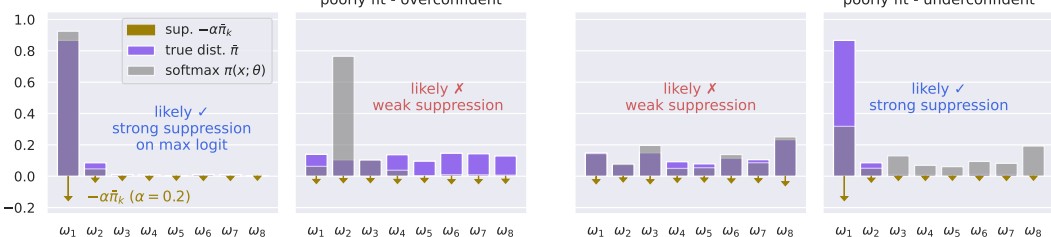

Figure 4: How the suppression gradient (Eq. (14)), *i.e.* the *difference* between LS and CE gradients, affects the logits differently. **LS affects the max logit differently depending on how well fit the model is for a given sample $x$.** In the left two when $U$ is lower (sharper softmax), the suppression on the max logit is *lower* when the model is poorly fit and likely to be wrong. In the right two when $U$ is higher (flatter softmax), the suppression is *higher* when the model is poorly fit and more likely to be correct. Thus, **LS degrades the softmax's ability to separate ✓ vs ✗ , hurting SC**.

which is the *difference* between the LS and CE gradients. This represents how LS influences training at the logit level in comparison to CE. Notably, it *only depends on the target $\bar{\pi}$*. Gradient descent involves updating weights in the *opposite* direction to the gradient. Hence $\alpha\bar{\pi}_k$ suppresses $v_k$ when the true probability $\bar{\pi}_k$ is higher. The second term $-\alpha/K$ uniformly increases the logits for all samples, which does not affect the softmax as it is invariant to uniform logit shifts $\pi(v) = \pi(v + \eta\mathbf{1}), \eta \in \mathbb{R}$.

## 4.2 IMBALANCED SUPPRESSION DEGRADES UNCERTAINTY RANKING OF ✓ VS ✗

We now consider how the suppression gradient affects the maximum logit $v_{\max}$,

$$\frac{\partial \mathcal{L}_{\text{sup}}^{\text{true}}}{\partial v_{\max}} = \alpha\bar{\pi}_{\hat{y}} - \alpha/K = \alpha[1 - P_{\text{error}}] - \alpha/K, \tag{15}$$

which shows that *the suppression on $v_{max}$ decreases as the probability of error increases*. This directly impacts softmax-based $U$ such as MSP (see Eqs. (1) and (12)), as the exponentiation of the softmax will be dominated by the largest logits, *especially* the max logit $v_{\max}$. The max logit will be especially dominant on high-confidence samples which are the last to be rejected.

Eq. (15) suggests that for predictions that share the same value of $U$, those with higher $P_{\text{error}}$ (more likely ✗), will have $v_{\max}$ suppressed *less*, whilst predictions with lower $P_{\text{error}}$ (more likely ✓) will have $v_{\max}$ suppressed *more*. Thus, softmax-based $U$ will have the *relative ranking* of correct ✓ and incorrect ✗ predictions degraded – less uncertain on ✗ , more uncertain on ✓. To further build an intuition, we consider how a model may not be uniformly well-fit over the data distribution during training. As illustrated in Fig. 4, varying levels of fit with regards to the true distribution $\bar{\pi}$ leads to differing $P_{\text{error}}$ for predictions with similar $U$, resulting in the degradation described above. Finally, as the suppression gradient in Eq. (15) is proportional to $\alpha$, this explains why stronger LS leads to greater degradation. **In summary: for a given predicted $U$ the $P_{\text{error}}$ will vary depending on how well fit the model is. LS suppresses the max logit *more(less)* when $P_{\text{error}}$ is *lower(higher)*, degrading the ranking of ✓ vs ✗ for softmax-based $U$, hurting SC.** We note that although our analysis is performed on *training* gradients, all our experiments are performed on *evaluation* data, suggesting that effects indeed generalise to unseen data. We also note that our explanation is purely based on the loss, and thus generalises across network architectures and tasks, matching the empirical results in Sec. 3.

**LS empirically leads to higher $v_{\max}$ on misclassifications ✗.** To further validate the effects of Eq. (15), we plot the mean±std. of $v_{\max}$ *given $\pi_{\max}$* for ResNet-50 on evaluation data in Fig. 5. We see that for LS the distribution of correct ✓ is *below* incorrect ✗, whilst for CE they are roughly similar. This provides further empirical support for Eq. (15) which states that when using LS, $v_{\max}$ is more strongly suppressed for lower $P_{\text{error}}$. For results on other models see Appendix B.2.

**LS empirically leads to increasing overconfidence on less-well-fit data.** To further investigate how LS affects uncertainty differently depending on how well-fit/knowledgeable the model is about data, we artificially introduce distribution-shifted data that we expect our models to be worse fit on. We use ImageNet-Sketch (Wang et al., 2019), a dataset containing sketches of each ImageNet class and evaluate on the *combination* of the 50,889 ImageNet-Sketch and 40,000 ImageNet evaluation images in Fig. 6. Even though the regularisation of LS improves the error rate, the degradation of

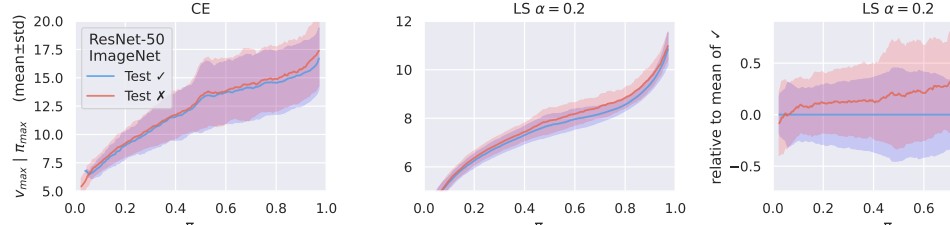

Figure 5: Distribution of the max logit $v_{\max}$ *given* the MSP $\pi_{\max}$ for correct ✓ and incorrect ✗ predictions on evaluation data. $v_{\max}$ is *lower* for ✓ for the LS model, whilst the distributions are roughly similar for CE. **This empirically matches the imbalanced max logit suppression described in Eq. (15)**. We calculate the mean±std. in a 0.05-wide sliding window.

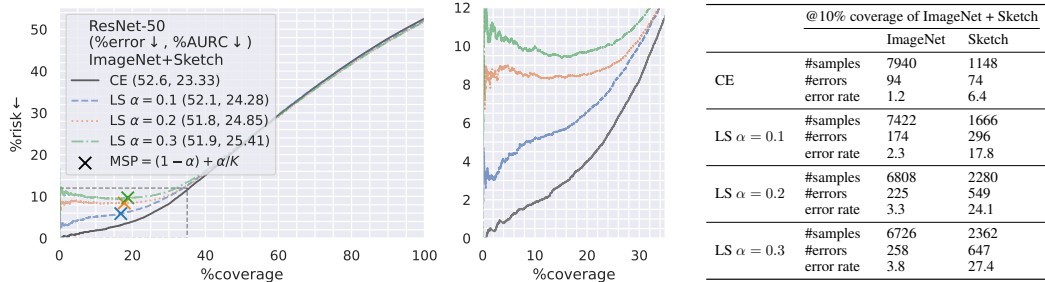

Figure 6: **Left**: Evaluation on the combination of ImageNet + ImageNet-sketch. We see that for low-uncertainty predictions, the degradation caused by LS is exacerbated when distribution shift is artificially introduced (vs Fig. 3). **Right**: statistics @10% coverage of the combined evaluation set. As the level of LS $\alpha$ increases, the number of accepted errors increases, *especially from ImageNet-Sketch*. This shows that **LS leads to increasing overconfidence on less-well-fit data**.

SC at lower coverages from LS is exacerbated by the distribution-shifted data. At 10% coverage of the combined evaluation set, LS leads to accepting many more errors compared to CE, with an increasing proportion originating from ImageNet-Sketch. That is to say, LS leads to increasing overconfidence on less-well-fit data (ImageNet-Sketch). This empirically highlights the situation illustrated on the *left* of Fig. 4 where confident and well-fit samples have their max logit suppressed, whilst confident but poorly fit samples do not, leading to samples with higher $P_{\text{error}}$ being relatively less uncertain (overconfident). For results on other models see Appendix B.2.

## 5 LOGIT NORMALISATION IMPROVES THE SC OF LS-TRAINED MODELS

Ideally, we would like to find a way to recover from the degradation caused by LS. A recent empirical study by Cattelan & Silva (2024) has shown that *logit normalisation* can improve the SC performance of many (but not all) pre-trained models. During deployment the logits are normalised by their $p$-norm and then the MSP score $\pi_{\max}$ is replaced by the normalised max logit,

$$U = -v'_{\max} = -\max_k v'_k, \quad \boldsymbol{v}' = [\boldsymbol{v} + s]/\|\boldsymbol{v} + s\|_p = [\boldsymbol{v} + s]/\left(\sum_k |v_k + s|^p\right)^{\frac{1}{p}}, \quad (16)$$

where $s$ is a scalar shift[5] and $p$ is found via AURC↓ grid search on a validation set. We investigate the efficacy of this approach, applying it to our LS-trained models. Figs. 7 and 13 show the SC performance with and without logit normalisation, and we indeed find that **applying logit normalisation greatly improves SC performance for models trained using LS**, allowing for improved error rate (@100% coverage) with good SC. This is further visualised for semantic segmentation in Fig. 8, where logit normalisation successfully mitigates the overconfident errors (bright yellow) caused by LS. However, we also find that logit normalisation does not notably

---

[5]We follow Cattelan & Silva (2024) and centralise the logits by subtracting the mean $s = -1/K \sum_k v_k$. For our experiments, this has little effect as $s \approx 0$, but it has other benefits (see Appendix F.3 for discussion).

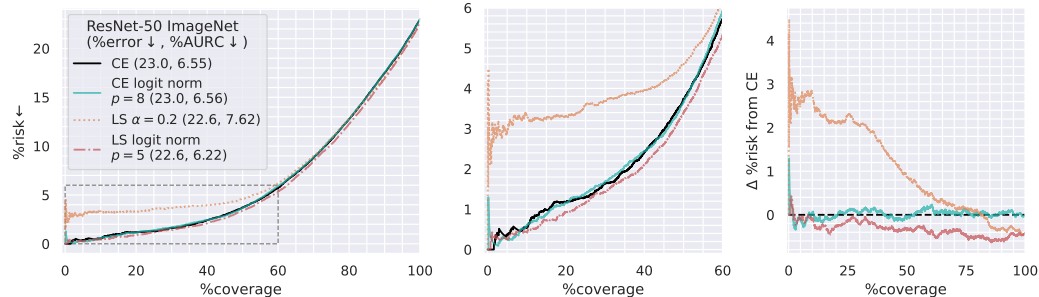

Figure 7: RC curves with inference-time logit normalisation. **Logit normalisation improves SC performance on LS models**, but has little effect on CE models. For other models, see Appendix B.2.

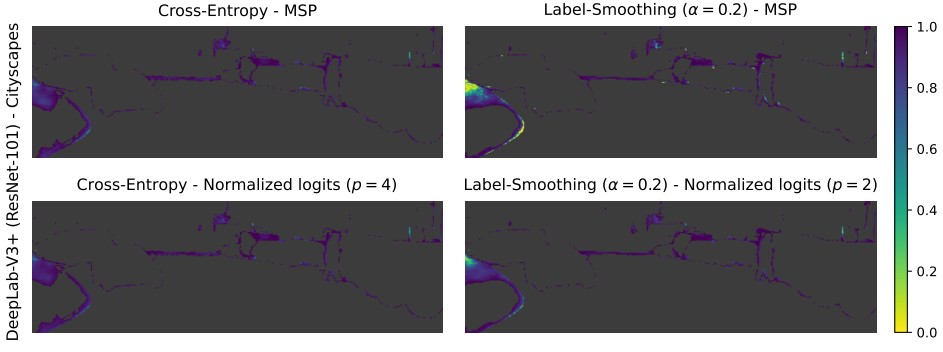

Figure 8: Visualisation of the effect of logit normalisation for segmentation (same scene as Fig. 1). **Logit normalisation significantly reduces the overconfidence of the LS-trained model**, although it has little effect on the uncertainty of the CE-trained model. We provide more figures in Appendix G.

improve the SC of CE models. This aligns with (Cattelan & Silva, 2024) where certain models do not benefit from logit normalisation so the authors suggest "falling back" to the MSP.

### 5.1 EXPLAINING THE EFFECTIVENESS OF LOGIT NORMALISATION.

Although Cattelan & Silva (2024) empirically validate this approach on a large number of pre-trained models, it remains unclear as to *why* it is so effective, or why it isn't effective sometimes (further discussion in Appendix F.4). Let us shed light on the interaction between logit normalisation and LS. If we examine Eq. (16), we see that it resembles the softmax, however, the logits are raised to power $p$ rather than exponentiated. Crucially, whilst the softmax is invariant to uniform shifts in the logits,

$$\pi_k(\boldsymbol{v}) = \frac{\exp v_k}{\sum_i \exp v_i} = \frac{\exp(v_k + \eta)}{\sum_i \exp(v_i + \eta)} = \pi_k(\boldsymbol{v} + \eta\mathbf{1}), \quad \forall \eta \in \mathbb{R} , \tag{17}$$

this is not the case for $\boldsymbol{v}'$. In fact, we have the following inequality for positive[6] logits $\boldsymbol{v}$:

**Result 1.** *For all vectors $\boldsymbol{v} \in (\mathbb{R}_{>0})^K$ with at least two different values and $p \in [1, +\infty[$, the ratio of the $\infty$-norm and $p$-norm strictly decreases when summing $\boldsymbol{v}$ with any uniform vector $\eta\mathbf{1}$, $\eta > 0$:*

$$v'_{max}(\boldsymbol{v}) = ||\boldsymbol{v}||_\infty / ||\boldsymbol{v}||_p \quad > \quad ||\boldsymbol{v} + \eta\mathbf{1}||_\infty / ||\boldsymbol{v} + \eta\mathbf{1}||_p = v'_{max}(\boldsymbol{v} + \eta\mathbf{1}) . \tag{18}$$

The proof can be found in Appendix E. Recalling that $||\boldsymbol{v}||_\infty = v_{max}$, Result 1 thus implies that for a given softmax output $\boldsymbol{\pi}(\boldsymbol{v}) = \boldsymbol{\pi}(\boldsymbol{v} + \eta\mathbf{1})$ and arbitrary corresponding $\boldsymbol{v}$, the greater the value of $\eta$ and thus $v_{max}$, the lower the value of the *normalised* max logit $v'_{max}$. That is to say **logit normalisation increases uncertainty when the max logit $v_{max}$ is higher for the same softmax probabilities $\pi$.**

Recall how in Sec. 4.2 we show how imbalanced logit suppression leads to degradation in SC. In particular Fig. 5 shows how LS leads to higher $v_{max}$ *given* $\pi_{max}$ for errors ✗ – this implies that, independent of the value of $\pi_{max}$, information about $P_{error}$ has been encoded in $v_{max}$. We see now that logit normalisation will increase the uncertainty (ranking) of errors ✗ relative to correct predictions

---

[6]Although this assumption may not necessarily hold, we find that in practice, $\pi_{max}$ and $v'_{max}$ are *dominated* by the larger *positive* logits, meaning the behaviour discussed still occurs empirically (Appendix F.3 and Fig. 22).

✓ using the information in $v_{\max}$, leading to improved SC for an LS model. That is to say **logit normalisation effectively reverses the effect of the imbalanced logit suppression from LS, improving SC**. We note that Cattelan & Silva (2024) find that models that are less confident (on both ✓ and ✗) tend to benefit more from logit normalisation, again pointing towards LS. Given our analysis and empirical results, **we strongly recommend logit normalisation for LS-trained models when performing SC**. On the other hand, Fig. 5 also shows that the distributions of $v_{\max}$ given $\pi_{\max}$ for correct ✓ and incorrect ✗ predictions are very similar for the CE model. This explains why for CE, logit-normalisation does not seem to help, as $v_{\max}$ does not provide useful information about $P_{\text{error}}$ given $\pi_{\max}$.

## 6 RELATED WORK

**Prediction with rejection.** Selective classification falls into the broader problem setting of *prediction with rejection*. In the case of SC, misclassifications are to be rejected (El-Yaniv & Wiener, 2010). The baseline approach is to use the MSP (Hendrycks & Gimpel, 2017; Geifman & El-Yaniv, 2017) and there have been a number of proposed training (Moon et al., 2020; Huang et al., 2020a; Ziyin et al., 2019; Zhu et al., 2024) and architectural (Geifman & El-Yaniv, 2019; Corbière et al., 2019) enhancements, however, recently the effectiveness of some of these enhancements has been called into question (Feng et al., 2023). Another scenario is out-of-distribution (OOD) detection (Yang et al., 2021; Hendrycks & Gimpel, 2017), where data from outside of the training distribution are to be rejected. There is a plethora of research in this field (Xia & Bouganis, 2022a; Sun et al., 2021; Liu et al., 2020; Zhang et al., 2023; Liu et al., 2023; Wang et al., 2022; Hendrycks et al., 2022). Recently, a combination of SC and OOD detection has been proposed, where the aim is to reject *both* misclassifications *and* OOD data (Jaeger et al., 2023; Xia & Bouganis, 2022b; Kim et al., 2023). Notably, Deep Ensembles (Lakshminarayanan et al., 2020) have arisen as a reliable method to improve performance in all three scenarios (Kim et al., 2023; Xia & Bouganis, 2023; Laurent et al., 2023; 2024). We believe, given the results of this work, that extending the investigation of LS (and other training enhancements) to other scenarios involving prediction with rejection is an important avenue of future work.

**Mixup.** Mixup (Zhang et al., 2018) and its variants (Yun et al., 2019; Franchi et al., 2021; Pinto et al., 2022; Liu et al., 2022d; Bouniot et al., 2023) are a set of regularisation techniques that involve interpolating between random pairs of samples at training time, modifying both inputs and targets. They are commonly found in many ImageNet training recipes (Tan & Le, 2021; Liu et al., 2021; 2022b) and are often used in conjunction with label smoothing (Wightman et al., 2021; Touvron et al., 2021; Liu et al., 2022c). Research into how Mixup affects SC is a particularly salient avenue of future work as (Zhu et al., 2024) also observe empirically that it can have a negative impact.

**Label smoothing.** Beyond prediction with rejection, LS is well-explored across various contexts. It has been shown to improve model calibration (Müller et al., 2019; Chun et al., 2020; Mukhoti et al., 2020; Liu et al., 2022a) as well as accuracy when training under label noise (Lukasik et al., 2020). Knowledge distillation (Hinton et al., 2015), where soft labels are provided by a teacher network, is also commonly linked and combined with LS (Müller et al., 2019; Gao et al., 2020; Yuan et al., 2020; Shen et al., 2021; Chandrasegaran et al., 2022) due to their similarity. Interestingly, in a similar vein to our work, pre-training using LS has been shown to harm transfer learning (Kornblith et al., 2021).

## 7 CONCLUDING REMARKS

In this work, we elucidate the effect of label smoothing (LS) on selective classification (SC). Our experiments across various tasks and architectures show that LS leads to consistent degradation in a model's ability to reject misclassifications, even if it improves accuracy. By analysing the logit-level gradients of the LS loss, we provide an explanation for this previously not understood behaviour – LS suppresses the max logit more(less) the more likely a prediction will be correct(wrong), degrading a model's uncertainty ranking of correct ✓ vs incorrect ✗ predictions. We then investigate post-hoc logit normalisation as a method to improve the degraded SC performance caused by LS. We find it to be highly effective and shed light on why – it reverses the effect of the aforementioned imbalanced suppression by increasing the uncertainty when the max logit is higher. We hope that our work encourages more research into understanding how different training techniques may impact model performance in downstream applications such as uncertainty estimation. A further discussion about the practical impact of our work, as well as potential future work can be found in Appendix H.

## ACKNOWLEDGMENTS

Guoxuan Xia's PhD is funded jointly by Arm and the EPSRC. This work was performed using HPC resources from GENCI-IDRIS (Grant 2023-[AD011011970R3]). We thank the anonymous reviewers for their valuable feedback.

## REPRODUCIBILITY STATEMENT

To ensure transparency, we use publicly available datasets: CIFAR-10/100, ImageNet, Cityscapes, IMDB, Bank Marketing, and Online shoppers. Our detailed experimental methods are outlined in Appendix C. Finally, the proof supporting our explanation of the effectiveness of logit normalisation is provided in Appendix E.

To help replicate our work, we share our source code on GitHub based on TorchUncertainty (Lafage et al., 2025),[7], including the configuration files and training code. We also release the most important models on Hugging Face.[8]

## ETHICS

Our primary objective is to contribute to enhancing the reliability of machine learning methods, with a particular focus on raising awareness about the negative effects of label-smoothing on selective classification. While our work aims to improve the robustness of ML systems, we acknowledge the potential risk that these advancements could be misapplied in harmful ways.

---

[7] https://github.com/ENSTA-U2IS-AI/Label-smoothing-Selective-classification-Code
[8] https://huggingface.co/ENSTA-U2IS/Label-smoothing-Selective-classification

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

## TABLE OF CONTENTS – SUPPLEMENTARY MATERIAL

# A    GLOSSARY OF NOTATION

We summarise the main notations used in the paper in Tab. 1.

Table 1: Glossary of Notation

| Notation | Meaning |
| --- | --- |
| $p(\cdot)$ | Probability density function |
| $P(\cdot)$ | Probability mass function |
| $\boldsymbol{x}$ | Input datum in $\mathbb{R}^D$ |
| $y$ | Label (categorical) from the set of $K$ possible labels $\{\omega_k\}_{k=1}^K$ |
| $p_{\text{data}}(\cdot)$ | True data distribution |
| $\mathcal{D}_{\text{tr}} = \{\boldsymbol{x}^{(n)}, y^{(n)}\}_{n=1}^N$ | Training dataset of inputs and labels drawn from $p_{\text{data}}(\boldsymbol{x}, y)$ |
| $\boldsymbol{\theta}$ | Model parameters |
| $\boldsymbol{v}$ | Logits (pre-softmax model outputs) in $\mathbb{R}^K$ |
| $v_{\max}$ | Maximum logit, $\max_k v_k(\boldsymbol{x}; \boldsymbol{\theta})$ |
| $\bar{\boldsymbol{\pi}}(\boldsymbol{x})$ | True categorical conditional data distribution, $\bar{\pi}_k = P_{\text{data}}(\omega_k|\boldsymbol{x})$ |
| $\boldsymbol{\pi}(\boldsymbol{x}; \boldsymbol{\theta})$ | Softmax output in $[0,1]^K$ of model, $\pi_k = \frac{\exp v_k}{\sum_i \exp v_i} = P(\omega_k|\boldsymbol{x}; \boldsymbol{\theta})$ |
| $(-)\text{MSP}/\pi_{\max}$ | Maximum softmax probability, $\max_k \pi_k(\boldsymbol{x}; \boldsymbol{\theta})$, negate for uncertainty |
| $\delta_{ij}$ | Kronecker delta, $\delta_{ij} = 1$ if $i = j$, and 0 if $i \neq j$ |
| $\text{KL}[\cdot||\cdot]$ | Kullback–Leibler divergence |
| $\mathcal{L}_{\text{CE}}(\boldsymbol{\theta})$ | Empirical cross entropy loss minimised over finite data $\mathcal{D}_{\text{tr}}$ |
| $\mathcal{L}_{\text{CE}}^{\text{true}}(\boldsymbol{\theta})$ | True cross entropy loss minimised over $p_{\text{data}}(\boldsymbol{x}, y)$. |
| $\alpha$ | Label smoothing parameter in $[0, 1]$ |
| $\mathcal{L}_{\text{LS}}(\boldsymbol{\theta}; \alpha)$ | Label smoothing loss |
| $\boldsymbol{x}^*$ | Test input datum |
| $f(\boldsymbol{x})$ | Classifier function |
| $\hat{y}$ | Label predicted by model for a given input, $\hat{y} = f(\boldsymbol{x}^*; \boldsymbol{\theta}) = \arg\max_\omega P(\omega|\boldsymbol{x}^*; \boldsymbol{\theta})$ |
| $\bar{\pi}_{\hat{y}}$ | True probability of label predicted by model $\bar{\pi}_{\hat{y}} = P_{\text{data}}(\hat{y}|\boldsymbol{x})$ |
| $P_{\text{error}}$ | Probability of a given label prediction being incorrect, $P_{\text{error}} = 1 - \bar{\pi}_{\hat{y}}$ |
| $U(\boldsymbol{x})$ | Scalar uncertainty score |
| $g(\boldsymbol{x}; \tau)$ | Binary rejection function, 0 (reject) if $U > \tau$ otherwise 1 (accept) |
| $\text{Risk}(f, g; \tau)$ | Average error on accepted samples for threshold $\tau$ |
| $\text{Coverage}(g; \tau)$ | Proportion of all data that is accepted for threshold $\tau$ |
| $\frac{\partial \mathcal{L}_{\text{sup}}}{\partial v_k}$ | Suppression logit gradient, difference between LS and CE gradients, $\frac{\mathcal{L}_{\text{LS}}}{\partial v_k} - \frac{\mathcal{L}_{\text{CE}}}{\partial v_k}$ |
| $\boldsymbol{v}'$ | $p$-normalised logits, $\boldsymbol{v}' = [\boldsymbol{v} + s] / \|\boldsymbol{v} + s\|_p$ with scalar shift $s$. |

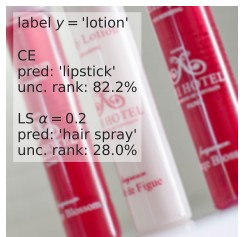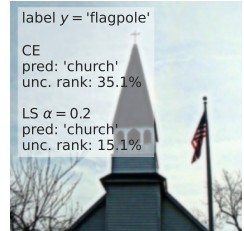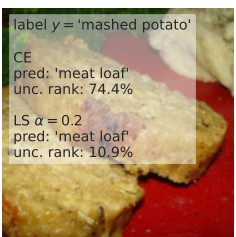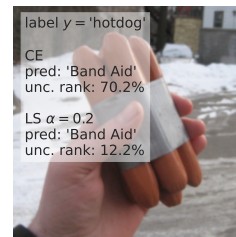

Figure 9: Illustrative examples of *overconfident* errors performed by our LS-trained ResNet-50 on evaluation data. Even though $P_{\text{error}}$ is high in all cases (*e.g.* due to multiple possible labels), the model predicts low (ranking) uncertainty. Note that even though the CE model is wrong as well, it has assigned higher ranking uncertainties, reflecting its superior SC ability shown in Fig. 3.

## B ADDITIONAL RESULTS

### B.1 ILLUSTRATIVE EXAMPLES OF IMAGENET OVERCONFIDENCE

Fig. 9 shows a few examples of overconfident misclassifications on the ImageNet evaluation data by our LS-trained ResNet-50.

### B.2 COMPLETE RESULTS FOR VIT AND DEEPLABV3+

We include additional experimental results that mirror those found in the main paper on ResNet-50, for ViT-S-16 and DeepLabV3+:

- Fig. 10 and Tab. 2 show full ResNet-50 and ViT-S-16 results on ImageNet + ImageNet-Sketch. We see that the behaviour of ViT-S-16 is similar to ResNet-50 (increasing numbers of confident errors from sketch as $\alpha$ increases), but less pronounced.

- Figs. 11 and 12 shows the distribution of $v_{\max}$ *given* $\pi_{\max}$ for ViT-S-16 and DeepLabV3+. We see that similarly to ResNet-50, for LS, the distribution of $v_{\max}$ is higher for errors ✗. Although it is less obvious, it is clear for both ViT-S-16 and DeepLabV3+ that for higher $\pi_{\max}$ the standard deviations overlap much less than for CE.

- Fig. 13 shows the effectiveness of logit normalisation for ViT-S-16 on ImageNet and DeepLabV3+ (ResNet-101) on Cityscapes. We also provide 2 additional segmentation figures in Appendix G.

Table 2: Statistics @10% coverage of the combined evaluation set. As the level of LS $\alpha$ increases, the number of errors increases, *especially the number of errors from ImageNet-Sketch*.

| ResNet-50 | | @10% coverage of ImageNet + Sketch | |
|---|---|---|---|
| | | ImageNet | Sketch |
| CE | #samples | 7940 | 1148 |
| | #errors | 94 | 74 |
| | error rate | 1.2 | 6.4 |
| LS $\alpha = 0.1$ | #samples | 7422 | 1666 |
| | #errors | 174 | 296 |
| | error rate | 2.3 | 17.8 |
| LS $\alpha = 0.2$ | #samples | 6808 | 2280 |
| | #errors | 225 | 549 |
| | error rate | 3.3 | 24.1 |
| LS $\alpha = 0.3$ | #samples | 6726 | 2362 |
| | #errors | 258 | 647 |
| | error rate | 3.8 | 27.4 |

| ViT-S-16 | | @10% coverage of ImageNet + Sketch | |
|---|---|---|---|
| | | ImageNet | Sketch |
| CE | #samples | 8529 | 559 |
| | #errors | 90 | 35 |
| | error rate | 1.1 | 6.3 |
| LS $\alpha = 0.1$ | #samples | 8229 | 859 |
| | #errors | 158 | 102 |
| | error rate | 1.9 | 11.9 |
| LS $\alpha = 0.2$ | #samples | 7927 | 1161 |
| | #errors | 196 | 161 |
| | error rate | 2.5 | 13.9 |
| LS $\alpha = 0.3$ | #samples | 7618 | 1470 |
| | #errors | 254 | 294 |
| | error rate | 3.3 | 20.0 |

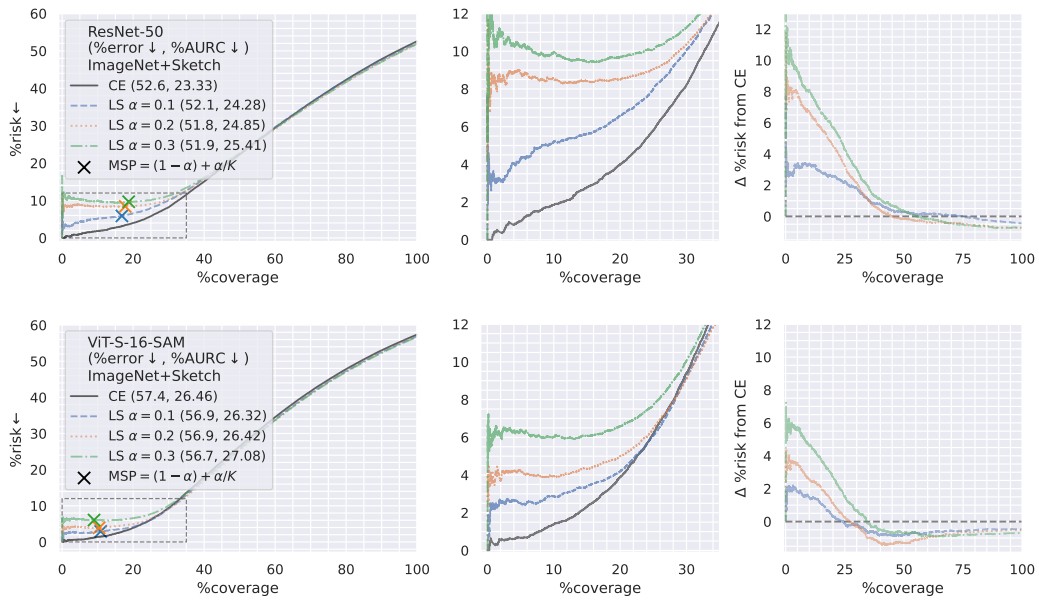

Figure 10: RC curves evaluating on the combination of ImageNet and ImageNet-sketch. For low-uncertainty predictions, the degradation caused by LS is exacerbated when distribution shift is artificially introduced. This shows that **LS leads to increasing overconfidence on less-well-fit data**.

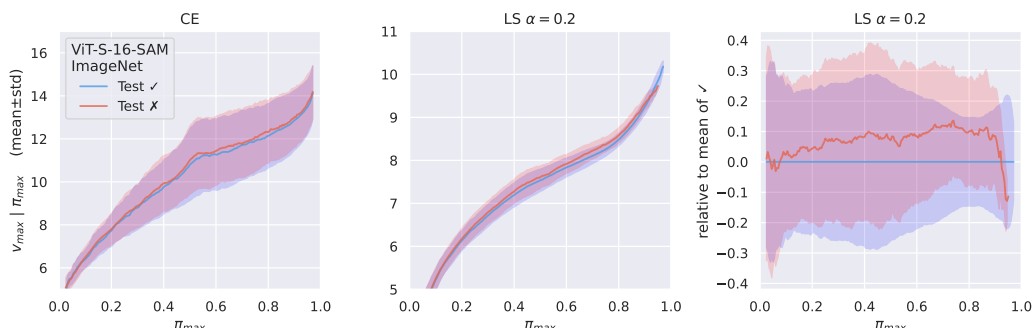

Figure 11: Distribution of the max logit $v_{max}$ *given* MSP $\pi_{max}$ for correct ✓ and incorrect ✗ predictions separately for ViT-S-16. Similarly to the main paper, the distribution of errors ✗ is higher than correct predictions ✓ for the LS-trained model. We calculate the mean±std. in a 0.05-wide sliding window.

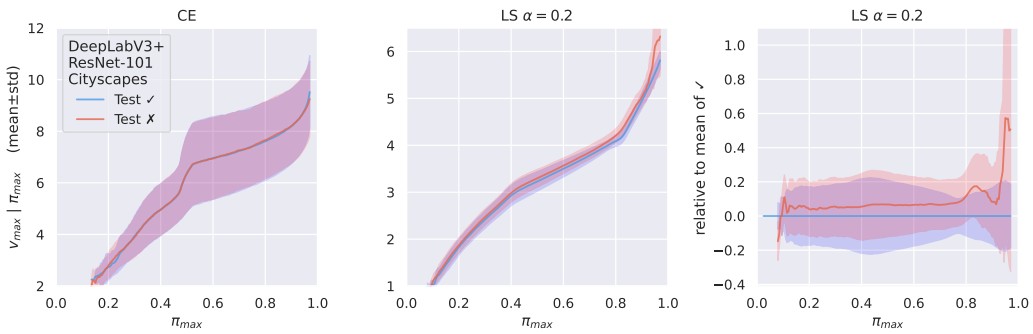

Figure 12: The same as Fig. 11 but for DeepLabV3+ (ResNet-101)

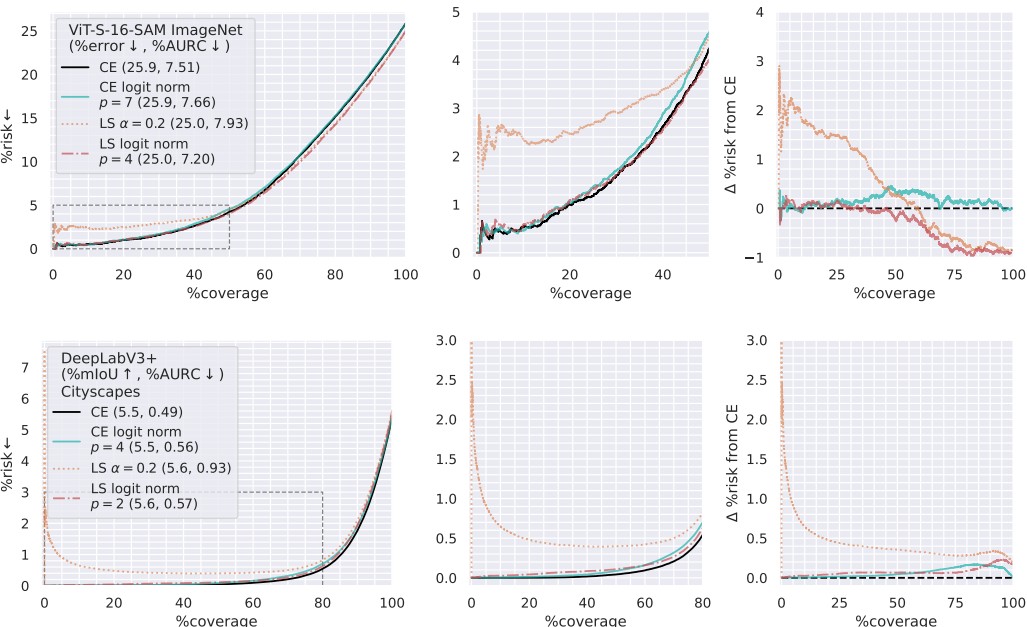

Figure 13: RC curve showing the effect of logit normalisation for ViT-S-16 and DeepLabV3+ (ResNet-101). The behaviour is similar to the results in the main paper. Logit normalisation is effective in improving the performance of the LS-trained models, bringing them close to the CE models. However, logit normalisation makes little difference to the CE-trained model.

### B.3 SMALL-SCALE CIFAR EXPERIMENTS

We also include experimental results on small-scale $32 \times 32$ CIFAR-100 (Krizhevsky, 2009). We train a DenseNet-BC (Huang et al., 2017) ($k = 12, L = 100$) to show further generality over model architecture families. Figs. 14 to 16 show results that mirror those found in the main paper for ResNet-50 on ImageNet, although we note that LS does not improve top-1 error rate in this case.

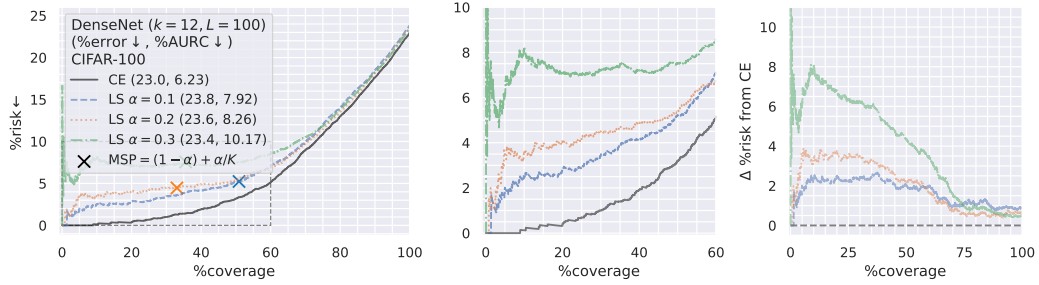

Figure 14: RC curves for DenseNet on CIFAR-100 – LS degrades SC

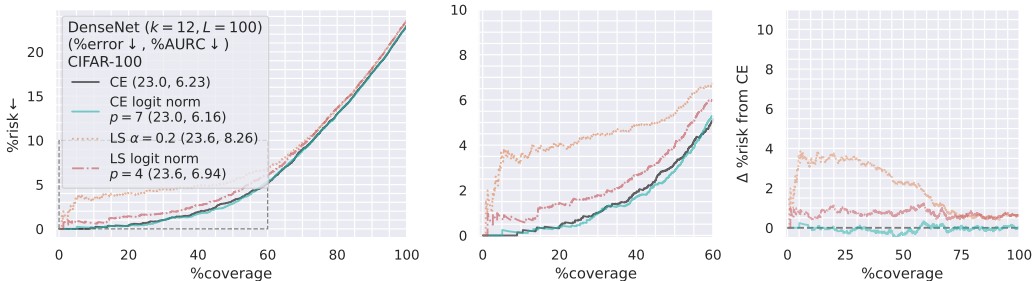

Figure 15: DenseNet on CIFAR-100 – logit normalisation improves SC for LS but not for CE

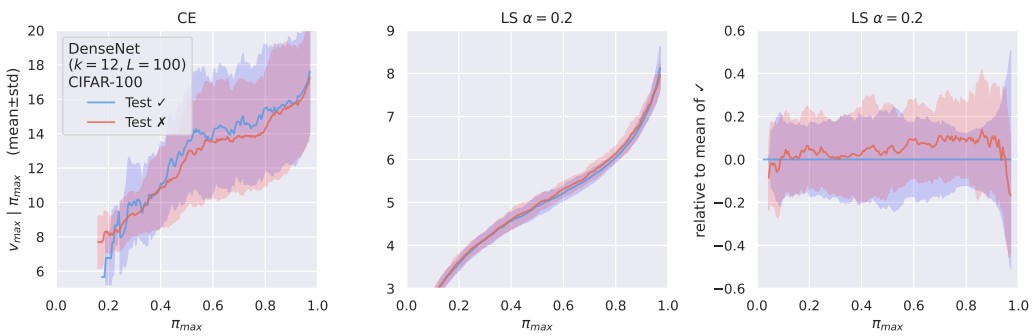

Figure 16: DenseNet on CIFAR-100 – LS leads to higher $v_{max}$ on errors ✗

### B.4 SMALL-SCALE TABULAR DATA EXPERIMENTS

To increase the scope of our experiments on the degradation of selective classification due to label-smoothing, we perform small-scale tabular binary classification. Following the setting of TabTransformer (Huang et al., 2020b), we train two-hidden-layer MLPs on Bank Marketing (Moro et al., 2014) and Online purchasing shoppers intentions (Sakar et al., 2019).

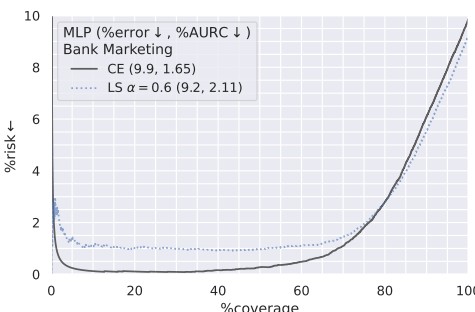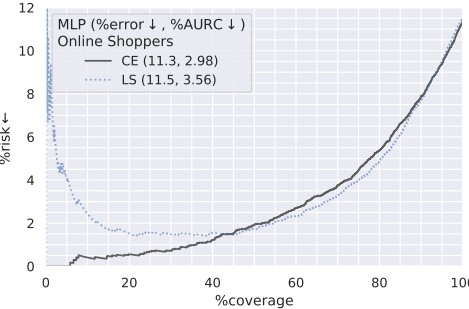

Figure 17: RC curves for MLPs trained on Bank Marketing and Online Shoppers using $\alpha = 0.6$, hence with a gap between the positive and negative smooth labels of 0.4.

In this binary classification setting, we set $\alpha$ to 0.6 to sufficiently reduce the gap between the optimal MSP corresponding to the positive and negative detections of the class at stake. With $\alpha = 0.6$ the minimum of the loss with label-smoothing is achieved for a prediction with a score of 0.7 when the hard label equals 1, and a score of 0.3 when the hard label is 0, resulting in a gap of a magnitude of 0.4. In Fig. 17, we see that in both cases, the AURC of the LS models are worse than those trained with the classical cross-entropy despite their accuracy being very similar. Furthermore, we see that the CE models are able to get much closer to *zero* risk, similar to the image classification experiments.

However, we note that the effect of label smoothing seems less pronounced for small-scale tabular data. We posit that this may be due to the data distributions being simpler and easier to capture, such that the neural networks are generally better fit and more knowledgeable about the data. Thus reducing the level of imbalanced suppression over training data (see Sec. 4). To address this lack of difficulty in fitting the distribution, we reduce the number of training points (as described in Appendix C).

Moreover, training small-scale and simple datasets drastically reduces the quality of our estimation of the Risk-Coverage curves. Given the limited number of data and the very high accuracy achieved by our MLPs, the number of errors used to estimate the risk is limited. The estimation of the "true" risk-coverage curve – which would be obtained with the whole distribution – is therefore imprecise and suffers from an important variability. The results may thus differ when starting the optimization process from different initializations, using a different batch composition and order, or due to the non-deterministic nature of some algorithms used to compute the backpropagation (Laurent et al., 2023).

### B.5 SMALL-SCALE NATURAL LANGUAGE DATA

With also provide small-scale natural language processing experiments training LSTM-based models (Sepp Hochreiter, Jürgen Schmidhuber , 1997) combined with two-layer perceptrons. We focus on LSTMs to add another architecture, and given that CNNs and transformers were already used in the image-classification setting (see e.g. Sec. 4).

We train two networks on the IMDB Movie Review dataset (Maas et al., 2011) with the classical binary cross-entropy loss and our implementation of the binary cross entropy with label-smoothing. Similarly to our results in image classification and tabular data classification, Fig. 18 shows that the LSTMs trained with label smoothing have worse AURC than models trained with cross-entropy despite their lower error-rate. Furthermore, the models also display greater error rates at high-confidence (low-coverage). We provide more extensive details on the training in Appendix C.

## C REPRODUCIBILITY

Alongside this document, we provide a code demo to train two ResNet-20 (He et al., 2016) on CIFAR-10 (Krizhevsky, 2009) with cross-entropy and label smoothing and compare the

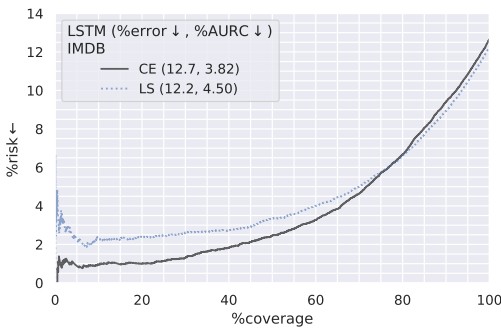

Figure 18: RC curves for LSTMs trained the IMDB Movie review dataset, using $\alpha = 0.6$, hence with a gap between the positive and negative smooth labels of 0.4.

corresponding Risk-Coverage curves. We recall that our code – based on TorchUncertainty (Lafage et al., 2025) – is available on GitHub.[9] We release the most important models on HuggingFace.[10]

Here, we provide the full details of our training recipes used in the main paper and in the additional results presented in Appendix B. All our models are trained with PyTorch (Paszke et al., 2019). We use the native implementation from the `CrossEntropyLoss` for label smoothing in the multi-class setting and a custom version of the `BCEWithLogitsLoss` for the binary experiments as the original does not support label smoothing. We use the original implementation of the authors for the negative label smoothing (Wei et al., 2022) experiments.

**NB**: The results presented in the final camera-ready ICLR 2025 version of this paper may differ slightly to earlier preprint versions. This is due to models being retrained with different random seeds, train-val-test splits, and having minor recipe and hardware differences after a consolidation of the experimental codebase to TorchUncertainty.[11] **Readers should use the information in this final version for reproducibility purposes**.

### C.1 IMAGE CLASSIFICATION

**DenseNet – CIFAR100.** For the DenseNet trained on CIFAR-100, we randomly split the original test set into 2000 images for validation and 8000 for evaluation. We take batches of 64 32×32-pixel images and train on a single GPU for 300 epochs using stochastic gradient descent and a starting learning rate of 0.1, Nesterov (Nesterov, 1983; Sutskever et al., 2013), a momentum of 0.9, and $1 \times 10^{-4}$ weight decay. We divide the learning rate by ten after 150 and 225 epochs. We use standard augmentations: we apply random crop with a four-pixel padding as well as random horizontal flip. We do not perform model selection and keep the last checkpoint.

**ViT-S-16 – ImageNet.** For our ViT-S-16 trained on ImageNet, we take batches of 2048 images and train on 4 A100s for 300 epochs with AdamW (Loshchilov & Hutter, 2019) with the $\beta$s equal to 0.9 and 0.999. We start with a linear warmup for 15 epochs, then use a cosine annealing scheduler with $3 \times 10^{-3}$ as the starting learning rate. The models are trained with non-adaptive sharpness-aware minimisation (SAM) (Foret et al., 2021; Chen et al., 2022) with $\rho = 0.2$. We use a dropout (Srivastava et al., 2014) rate of 0.1 but no attention dropout. We transform the training images with a standard random resized crop to 224×224 pixels using bicubic interpolation and a random horizontal flip. For evaluation, we center-crop the images to this resolution. For ImageNet, we do not perform model selection and keep the last checkpoint. However, we randomly extract a validation set of 10,000 images from the validation set to perform logit normalisation, and evaluation on the remaining 40,000.

---

[9] https://github.com/ENSTA-U2IS-AI/Label-smoothing-Selective-classification-Code

[10] https://huggingface.co/ENSTA-U2IS/Label-smoothing-Selective-classification

[11] https://torch-uncertainty.github.io/

**ResNet-50 – ImageNet.** Our ResNet-50 is trained on 4 A100 with stochastic gradient descent for 120 epochs using a batch size of 1024 images. After five epochs of linear warmup, we use a cosine annealing scheduler starting with a learning rate of 0.4 with a momentum of 0.9 and a weight decay of $1 \times 10^{-4}$. We use the same transformation of the images as for the ViT and select the last checkpoint for inference. We use the same validation set as for the ViT for logit normalisation.

### C.2 SEMANTIC SEGMENTATION

**Deeplabv3+.** We train a Deeplabv3+ on CityScapes (Cordts et al., 2016) with a ResNet-101 (He et al., 2016) backbone pre-trained on ImageNet (Russakovsky et al., 2015). We use stochastic gradient descent with a base learning rate of 0.01, divided by 10 for the backbone weights, and reduced following the "poly" policy (Liu et al., 2015) with a power of 0.9. The weights are optimised with a momentum of 0.9 and a weight decay of $10^{-4}$. We take a batch size of 16 images and train for 30,000 steps. During training, we randomly crop the input images and targets to squares with 768-pixel-long sides. We apply random horizontal flip and colour-jitter with the classical parameters: brightness, contrast, and saturation levels of 0.5. For testing, we use the images at their original resolution and do not perform any test time augmentations. For the RC curves, we randomly sample 5000 predictions per image extracted prior to the final interpolation to compute the coverage and error rates. We keep the pixel-wise locations of the samples when changing the level of label smoothing $\alpha$ to ensure fair comparisons.

### C.3 TABULAR DATA

We perform the tabular data binary classification on two UCI datasets: Bank Marketing and Online shoppers. These two datasets have an input dimension and a number of samples of 7 and 45,211 for the former and 16 and 12,330 for the latter. In both cases, we select 80% of the data points for the test set and train the models for 10 epochs using PyTorch's default Adam optimizer and a batch size of 128. Similarly to the other experiments of this paper, we do not perform model selection and keep the last checkpoint. Our models are MLPs with two hidden layers, whose size depends on the input size as follows: the first hidden layer has 4 times as many neurons as the dimension of the input data, and the second has half the number of neurons of the first layer.

### C.4 NATURAL LANGUAGE DATA

For the training on the IMDB (Maas et al., 2011) dataset, we tokenize the data with `nltk` and convert them to 300-dimensional vectors using GloVe 840B (Pennington et al., 2014). The architecture of the models is defined as follows:

- an LSTM layer with a 300 input dimension and a 256 output dimension, followed by a dropout layer of rate 0.2,

- a linear layer keeping the dimension of the vectors unchanged, on which we apply ReLU, followed by a second dropout layer with the same rate,

- two linear layers with ReLU reducing the dimension to 128 and 1, respectively (binary classification).

Finally, we train the models with PyTorch's Adam default for 10 epochs using the pre-made train test split. The code is available on GitHub.

### C.5 LOGIT CALCULATIONS

We perform logit evaluation calculations in *double precision* (softmax, logit normalisation, entropy etc.), in order to reduce the impact of numerical error.

We perform normalisation as in Eq. (16) on logits $v$. The value of $p$ is searched over $\{1,2,3,4,5,6,7,8\}$ on the corresponding validation data with the optimisation metric as AURC↓. We set $s = -1/K \sum_k v_k$ following Cattelan & Silva (2024) (see Appendix F.3 for a discussion on this choice).

## D ANALYSIS OF EMPIRICAL LOSS (ONE-HOT LABELS)

We can perform a similar analysis as in Sec. 3 using the empirical loss (Eqs. (2) and (6)) rather than $\mathcal{L}^{\text{true}}$ as in the main paper. We leave this to the Appendix as the conclusions are similar to analysing $\mathcal{L}^{\text{true}}$, however, the presence of one-hot targets makes it less convenient to reason about the probability of error $P_{\text{error}}$ and how well fit the model is to the true conditional distribution $P_{\text{data}}(y|\boldsymbol{x})$ for a given data sample. Taking the gradients of Eqs. (2) and (6) as in Eq. (13),

$$\frac{\partial \mathcal{L}_{\text{CE}}}{\partial v_k} = -\left[\delta_{y\omega_k} - \pi_k\right], \quad \frac{\partial \mathcal{L}_{\text{LS}}}{\partial v_k} = -\left[\Big[\underbrace{(1-\alpha)\delta_{y\omega_k}}_{\text{data supervision}} + \underbrace{\alpha/K}_{\text{regularisation}}\Big] - \pi_k\right], \tag{19}$$

which in turn gives the suppression gradient,

$$\frac{\partial \mathcal{L}_{\text{sup}}}{\partial v_k} = \frac{\partial (\mathcal{L}_{\text{LS}} - \mathcal{L}_{\text{CE}})}{\partial v_k} = \frac{\partial \mathcal{L}_{\text{LS}}}{\partial v_k} - \frac{\partial \mathcal{L}_{\text{CE}}}{\partial v_k} = \alpha\left[\delta_{y\omega_k} - 1/K\right] , \tag{20}$$

which is once again independent of the model output $\boldsymbol{\pi}$. In this case, as the label $y$ has already been sampled, the model is either right ✓ or wrong ✗, giving two different suppression gradients for the max logit $v_{\text{max}}$,

$$\frac{\partial \mathcal{L}_{\text{sup}}^{✓}}{\partial v_{\text{max}}} = \alpha\left[1 - 1/K\right] = \alpha - \alpha/K, \quad \frac{\partial \mathcal{L}_{\text{sup}}^{✗}}{\partial v_{\text{max}}} = \alpha\left[0 - 1/K\right] = -\alpha/K . \tag{21}$$

We can see that the max logit is more strongly suppressed during training when the prediction is correct, which aligns with the analysis of $\mathcal{L}^{\text{true}}$ and $P_{\text{error}}$ in the main paper. **When the model is correct during training, the max logit is suppressed, but when it is incorrect the max logit is not suppressed. Thus the uncertainty ranking of correct ✓ vs incorrect ✗ data samples is degraded, harming SC.** This leads us to the same conclusions as in Sec. 4 and also aligns with the behaviour in Fig. 5 (right) where $v_{\text{max}}$ is higher for errors given the value of MSP.

## E RESULT AND PROOF

**Result.** *For all strictly positive vectors $\boldsymbol{v} \in (\mathbb{R}_{>0})^K$ containing at least two different values and $p \in [1, +\infty[$, the ratio of the infinite norm and the p-norm strictly decreases when summing $v$ and any uniform vector $\eta\mathbf{1}$, $\eta$ strictly positive:*

$$\frac{||\boldsymbol{v}||_\infty}{||\boldsymbol{v}||_p} > \frac{||\boldsymbol{v} + \eta\mathbf{1}||_\infty}{||\boldsymbol{v} + \eta\mathbf{1}||_p}. \tag{22}$$

*Proof.* Let there be a real $\eta > 0$. Take $p \geq 1$ the dimension of the norm and $\boldsymbol{v}$ a vector of dimension $K \geq 1$ of strictly positive elements $v_k$ for $1 \leq k \leq K$, such that there exists $1 \leq i \leq K$ such that $v_i < \max_{k \leq K} v_k$. We have that

$$1 + \frac{\eta}{v_k} \geq 1 + \frac{\eta}{\max_{k \leq K} v_k}, \tag{23}$$

and, for at least $i$, we have the same equation, yet with strict inequality. We can adapt Eq. (23) to get

$$v_k + \eta \geq \frac{\max_{k \leq K}(v_k + \eta)}{\max_{k \leq K} v_k} v_k. \tag{24}$$

And when set to exponent $p \geq 1$, we obtain

$$\frac{(v_k + \eta)^p}{\max_{k \leq K}(v_k + \eta)^p} \geq \frac{v_k^p}{\max_{k \leq K} v_k^p}. \tag{25}$$

Similar to Eq. (23), please note that using $k = i$, we get the same equation as Eq. (25), although with a strict inequality. We can now sum on the elements of $\boldsymbol{v}$ to get

$$\frac{\sum_{k=1}^{K}(v_k + \eta)^p}{\max_{k \leq K}(v_k + \eta)^p} > \frac{\sum_{k=1}^{n} v_k^p}{\max_{k \leq K} v_k^p}. \tag{26}$$

By taking the inverse (all values are strictly positive), we get

$$\frac{\max\limits_{k \leq K} v_k^p}{\sum\limits_{k=1}^{n} v_k^p} > \frac{\max\limits_{k \leq K} (v_k + \eta)^p}{\sum\limits_{k=1}^{K} (v_k + \eta)^p}. \tag{27}$$

And setting the equation to the exponent $p^{-1}$ and replacing the maxima of the $v_k$ and $v_k + \eta$ with the infinite norm, $||\boldsymbol{v}||_\infty$ and $||\boldsymbol{v} + \eta\mathbf{1}||_\infty$ respectively, we obtain the result. $\qquad\square$

## F    DISCUSSIONS

### F.1    $U$ OTHER THAN MSP

In the main body of the paper we focus solely on MSP as our uncertainty score $U$. Fig. 19 shows how LS affects the SC behaviour of MSP (top) compared two other softmax scores: DOCTOR (Granese et al., 2021) ($U = -||\boldsymbol{\pi}||_2$) (middle) and entropy ($U = H(\boldsymbol{\pi}) = -\sum_k \pi_k \log \pi_k$) (bottom). We see that the behaviour is very similar across all three softmax scores, with increasing the level of LS $\alpha$ degrading all the scores. As MSP is the (marginally) best performing and the most commonly used, we thus choose to focus on it for the main paper.

Fig. 20 shows the SC performance of Energy ($U = -\log \sum_k \exp v_k$) (Liu et al., 2020), a popular OOD detection score, compared to MSP with and without LS. We see that Energy performs much worse than MSP at SC. This aligns with a large body of existing work (Xia & Bouganis, 2022b; Jaeger et al., 2023; Kim et al., 2023; Yang et al., 2024; Zhu et al., 2024) that empirically finds that uncertainty scores designed for OOD detection perform poorly at detecting misclassifications and are thus not suitable for SC. Thus we choose not to investigate any OOD detection scores in the main paper. We remark that LS also has a strong negative effect on the SC performance of Energy. This is unsurprising as Energy is dominated by the max logit. An important future direction would be to investigate the effect of LS on various OOD detection scores on the task of OOD detection.

Finally, we also choose to omit various training/architecture-based approaches such as (Ziyin et al., 2019; Geifman & El-Yaniv, 2019; Moon et al., 2020; Zhu et al., 2024), as we aim to focus our investigation solely on understanding the training effects of LS. We note that LS (potentially combined with post-hoc logit normalisation) is considerably simpler to implement than the aforementioned training-based approaches, and has been shown to be stable in many more use cases and so may be more likely to be chosen by a practitioner. Besides, recent work (Feng et al., 2023) has suggested that MSP applied to the methods in (Ziyin et al., 2019; Geifman & El-Yaniv, 2019) actually performs better than the $U$s proposed in them (reject logit/selection head).

### F.2    ALEATORIC AND EPISTEMIC UNCERTAINTY

We recognise that the conceptual decomposition of predictive uncertainty into *aleatoric* and *epistemic* uncertainty (Gal, 2016; Hüllermeier & Waegeman, 2021; Kirsch, 2024) can be applied to the discussion throughout this paper, and that some readers may be confused as to why we do not use these terms. We choose to use simpler, more direct language as we believe it more efficiently conveys our discussion, reduces the number of concepts to introduce and also reduces any potential confusion.

### F.3    ON THE ASSUMPTION THAT LOGITS ARE POSITIVE

**Largest positive logits dominate.**    Result 1 assumes that all elements in the logit vector are $> 0$, which is not necessarily true. However, we also find empirically that both $v'_{\max}$ and $\pi_{\max}$ tend to be dominated by the largest *positive* logits. This is intuitive as exponentiating or raising to power $p > 1$ will amplify the larger logits. This is shown in Fig. 21, where we plot the mean±std of $v, v^5$ and $\exp v$ for the sorted logits of ResNet-50 $\alpha = 0.2$ on the ImageNet evaluation set ($p = 5$ is optimal on the validation data for logit normalisation in this case). **Thus $\pi_{\max}(\boldsymbol{v}) \approx \pi_{\max}(\boldsymbol{v}_{\text{top-}k})$ and $v'_{\max}(\boldsymbol{v}) \approx v'_{\max}(\boldsymbol{v}_{\text{top-}k})$, where the top-$k$ logits are positive and dominate the computation. As Result 1 holds for $v'_{\max}(\boldsymbol{v}_{\text{top-}k})$, we still expect logit normalisation to increase uncertainty for higher $v_{\max}$ when comparing samples with similar $\pi_{\max}$, even if not all $v_k > 0$.** (Note that we omit shift $s = -1/K \sum_k v_k$ as it is $\approx 0$ in our experiments).

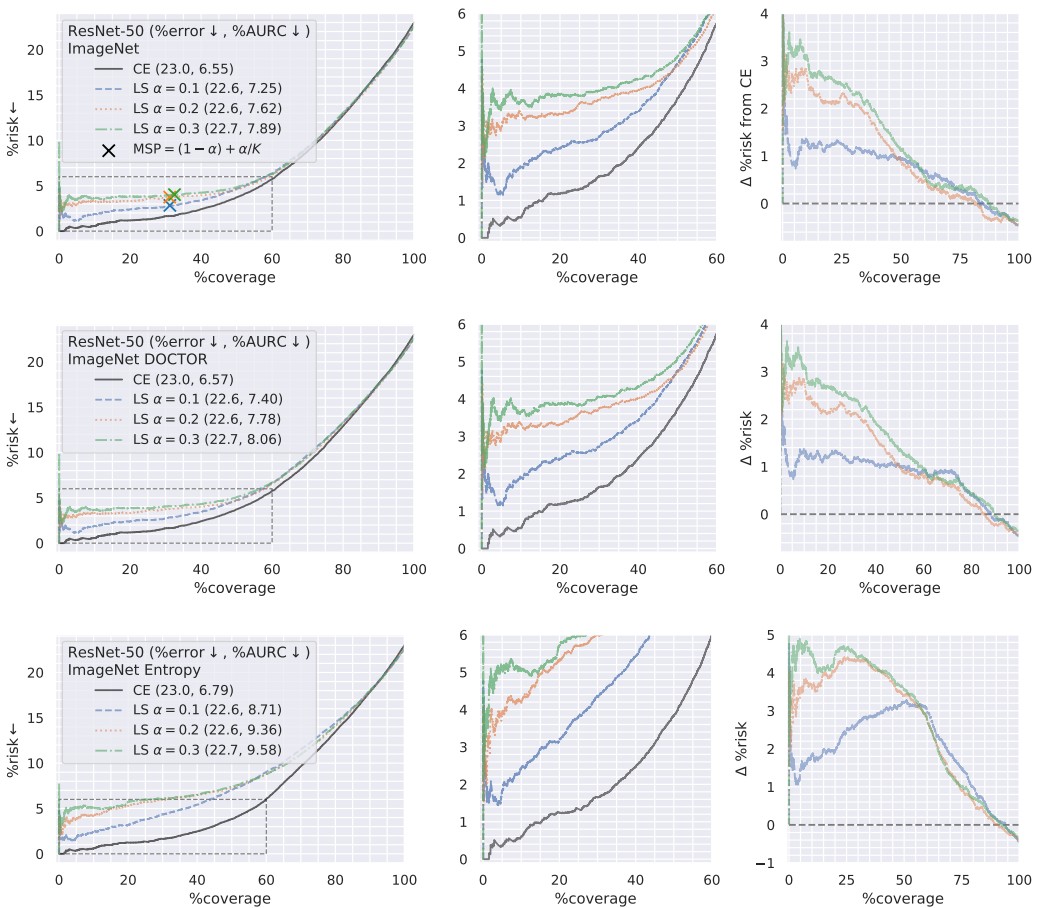

Figure 19: The effect of LS on different softmax scores: MSP (top), DOCTOR (middle), Entropy (bottom). We see that the behaviour is very similar, with MSP being the best performer.

We can also consider the scenario where we add $\eta$ to only the top-$k$ logits. This more aptly describes the empirical logit behaviour compared to adding $\eta$ to all logits, as the values of the lower ranking logits vary much less than the higher ranking ones (Fig. 21). Here we would expect $\pi_{\max}(\boldsymbol{v})$ to increase very slightly, but would expect $v'_{\max}(\boldsymbol{v})$ to decrease as the numerator of Eq. (26) would be dominated by the top-$k$ largest logits.

Fig. 22 shows how *empirically* logit normalisation indeed increases uncertainty for higher $v_{\max}$. We plot the mean±std of $v'_{\max}$ given $v_{\max}$ for samples in different MSP bins. We see clearly that in almost all cases, for samples with similar $\pi_{\max}$, the normalised max logit $v'_{\max}$ *decreases* as the original max logit $v_{\max}$ *increases*. As shown in Figs. 5, 11, 12 and 16, LS leads to misclassifications ✗ having higher $v_{\max}$ than correct predictions ✓. Thus, logit normalisation is able to improve the SC performance of LS-trained models by penalising the confidence of higher max logit values.

**Logit centralisation.** We follow Cattelan & Silva (2024) and suggest to shift the logits by their mean during normalisation,

$$\boldsymbol{v}' = \frac{\boldsymbol{v} - \mu(\boldsymbol{v})}{\|\boldsymbol{v} - \mu(\boldsymbol{v})\|_p}, \quad \mu(\boldsymbol{v}) = \frac{1}{K}\sum_k v_k. \tag{28}$$

In our experiments, this has little-to-no effect as $\mu \approx 0$ (for example for ResNet-50 $\alpha = 0.2$ on the ImageNet evaluation data $\mu$ averages $\sim 3 \times 10^{-3}$ with std. $\sim 1 \times 10^{-3}$). This is corroborated by Cattelan & Silva (2024)'s observations in their Appendix G, where they find this shift does not affect the vast majority of their models.

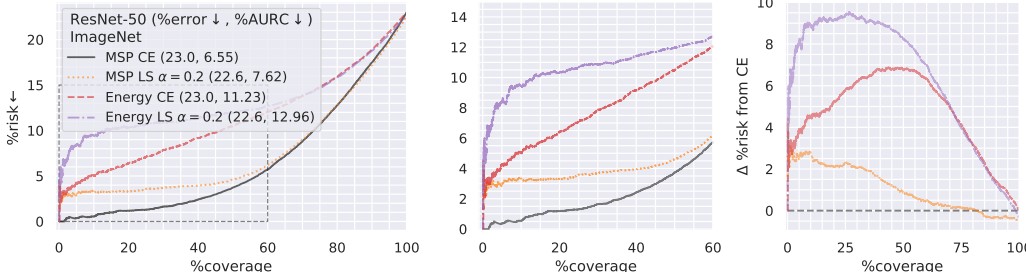

Figure 20: SC performance of MSP and Energy (OOD detection score). Energy significantly underperforms MSP. This behaviour is in line with existing work that shows that uncertainty scores designed for OOD detection are not suitable for SC.

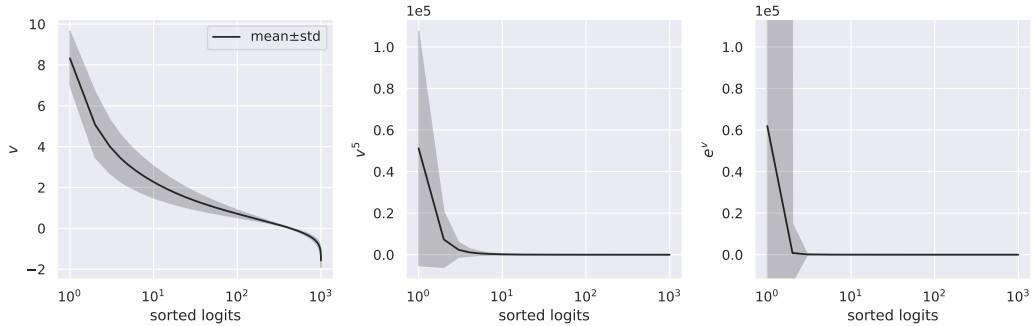

Figure 21: Mean±std after logits have been sorted highest to lowest for ResNet-50 $\alpha = 0.2$ on the ImageNet evaluation set. We see that $v^5$ and $\exp v$ are much larger for the top $< 10$ logits. Thus these logits dominate $\pi_{\max}$ and $v'_{\max}$.

We note that according to the analysis in Sec. 5.1 and Appendix F.3, for logit normalisation to be effective, $v'_{\max}$ and $\pi_{\max}$ should be dominated by positive logits. Logits are not necessarily constrained to be positive. They can be arbitrarily shifted without affecting the cross entropy loss (*e.g.* via bias $\boldsymbol{b}$ of the final layer), or derived as log probabilities which are necessarily negative. Thus we suggest applying the above shift to ensure that the largest logits are positive. We note that other shifts are possible (for example $s = -\min_k v_k$) to achieve the same purpose but we leave the exploration of this to future work.

### F.4 EXPLANATIONS OF LOGIT NORMALISATION IN (CATTELAN & SILVA, 2024)

The work that introduces logit normalisation (Cattelan & Silva, 2024) is primarily an experimental study, where the focus is on extracting empirical takeaways. However, the authors do discuss potential reasons for the effectiveness of the approach in their Appendix B. They observe that models that are generally (on both ✓ and ✗) more uncertain benefit from logit normalisation and so suggest that logit normalisation alleviates "underconfidence" by reducing the influence of lower-ranked logits on the uncertainty of a prediction (using analysis similar to ours in Appendix E). We believe their explanation is ultimately incomplete as:

1. They do not clearly delineate the definition of over/underconfidence used in model calibration with that used in selective classification. This is an issue since calibration is concerned with absolute marginal (averaged over data samples) properties, whilst selective classification is concerned with relative conditional (per sample) properties. It is possible to be extremely over/underconfident in the calibration sense, whilst being optimal for SC, or very well calibrated and worse at distinguishing correct vs incorrect samples (Zhu et al., 2024).

2. Although they elucidate some of the mechanics of logit normalisation, showing that logit normalisation reduces the impact of smaller logits on uncertainty, they do not link it to

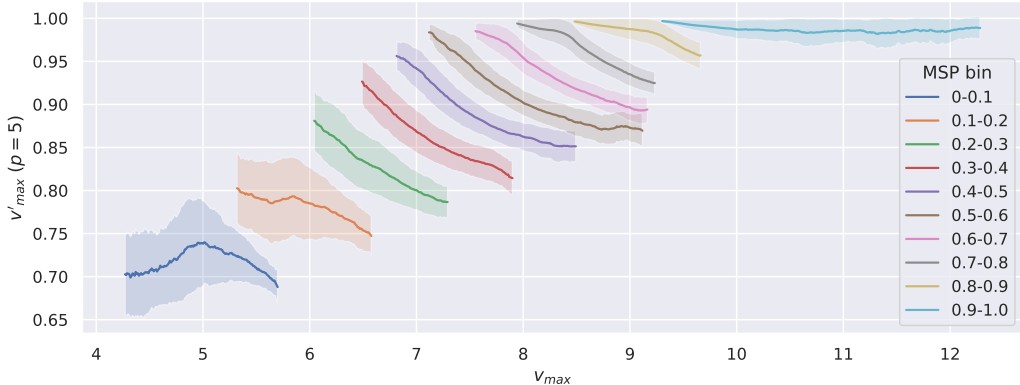

Figure 22: $v'_{\max}$ given $v_{\max}$ within bins of similar MSP for ResNet-50 $\alpha = 0.2$ on the ImageNet evaluation set. Generally, $v'_{\max}$ decreases as $v_{\max}$ increases, **showing empirically how logit normalisation increases uncertainty for higher $v_{\max}$**. We calculate the mean±std. of $v'_{\max}$ in a 0.2-wide sliding window for samples with $v_{\max}$ within mean±2std of $v_{\max}$ within the bin (to remove noisy averages).

> any observed model behaviour that differs between correct ✓ vs incorrect ✗ predictions, instead just speculating that models may be underconfident on certain sets of samples. To explain why an approach improves SC, we need to know how it treats ✓ and ✗ *differently* to improve the rank ordering of uncertainties between ✓ and ✗.

On the other hand, our explanation focuses solely on the relative ranking of uncertainties between ✓ vs ✗ samples and is able to link the mechanics of logit normalisation to how the behaviour of correct ✓ vs incorrect ✗ predictions differ under label smoothing (Fig. 5 and Result 1).

### F.5 ANALYSIS OF NEGATIVE LABEL SMOOTHING (WEI ET AL., 2022)

We consider the framework developed in the main paper to study Generalized Label Smoothing (GLS) with negative smoothing values – called Negative Label Smoothing (NLS) – as suggested by Wei et al. (2022). The definition of GLS is the same as the original from Szegedy et al. (2016), except for the domain of the label smoothing value $\alpha$, which can take values in $(-\infty, 1]$. Wei et al. (2022) suggest that using negative values for the label smoothing value can lead to improved performance when training with noisy labels as well as in the "clean" setting. It is tempting to consider, as negative $\alpha$ ought to reverse the logit suppression described in Eq. (15). However, here we show that NLS can lead to unstable training characteristics.

The per-sample cross-entropy between a single softmax prediction $\pi$ and the corresponding true categorical distribution $\bar{\pi}$ is convex and can be written as follows:

$$\text{CE}(\pi, \bar{\pi}) = - \sum_{k \in [\![1,K]\!]} \left[ (1-\alpha)\bar{\pi}_k + \frac{\alpha}{K}\mathbf{1} \right] \log \pi_k. \tag{29}$$

For $\alpha \geq 0$, which covers vanilla cross entropy ($\alpha = 0$) and regular label smoothing ($\alpha > 0$) a single global minimum is attained in $\pi = (1-\alpha)\bar{\pi} + \frac{\alpha}{K}\mathbf{1}$. Here, *the gradient of the loss is zero* therefore creating a balance and stabilizing training, taking the gradient as in Eq. (13),

$$\frac{\partial \text{CE}(\pi, \bar{\pi})}{\partial v_k} = - \left[ \underbrace{\left[ (1-\alpha)\bar{\pi}_k + \alpha/K \right]}_{\text{target}} - \underbrace{\pi_k}_{\text{softmax output}} \right]. \tag{30}$$

However, in the case of a negative label smoothing value $\alpha < 0$ the target can be *outside* $[0,1]$, but the softmax output is constrained to $[0,1]$, thus $\pi = (1-\alpha)\bar{\pi} + \frac{\alpha}{K}\mathbf{1}$ is not achievable and *the gradient will always have some magnitude* and *there is no optimisation minimum*.

Indeed, if we consider for one-hot $\bar{\pi}$, the $i$-th terms s.t. $\bar{\pi}_i = 0$ of Eq. (29) will be strictly negative with a negative-label-smoothed target. Since the log of the softmax probabilities $\pi$ can take values

in $(-\infty, 0]$, so can the loss. Being convex, there cannot exist a local minimum, and therefore, *the gradient can never be zero* and pushes its value to $-\infty$ by reducing the value of the corresponding logit to $-\infty$ and pushing the others to $\infty$. We hypothesise that this unbalance explains the instability of the method that users reported on the GitHub repository of the original paper. Unfortunately, we too were not able to reproduce the experiments of Wei et al. (2022) and obtain stable enough training runs to perform experiments on NLS and confirm this theoretical analysis.

## F.6 EXISTING BENCHMARKS AND TRAINING RECIPES WITH LS

Although we do not exhaustively search all training recipes for all models benchmarked in (Galil et al., 2023; Cattelan & Silva, 2024), we do provide a number of examples of evaluated models trained with label smoothing. We also provide links to publicly available training repositories, as not all papers mention label smoothing even when it is used in training. Upon inspection of (Galil et al., 2023; Cattelan & Silva, 2024), these models do in fact seem to underperform at selective classification (and Cattelan & Silva (2024) report that their AURCs benefit from logit normalisation).

- EfficientNet (Tan & Le, 2019): `https://github.com/tensorflow/tpu/blob/master/models/official/efficientnet/main.py#L249`
- EfficientNet-V2 (Tan & Le, 2021): `https://github.com/google/automl/blob/master/efficientnetv2/datasets.py#L658`
- DeiT (Touvron et al., 2021): `https://github.com/facebookresearch/deit/blob/main/main.py#L101`
- Swin-Transformer (+V2) (Liu et al., 2022b; 2021): `https://github.com/microsoft/Swin-Transformer/blob/main/config.py#L70`
- ConvNeXt (Liu et al., 2022c): `https://github.com/facebookresearch/ConvNeXt/blob/main/main.py#L105`
- Torchvision (Paszke et al., 2019) (various): `https://github.com/pytorch/vision/tree/main/references/classification`

Galil et al. (2023) state that some of their best performing (at SC) ViT models (Dosovitskiy et al., 2021; Steiner et al., 2022; Chen et al., 2022) are trained with label smoothing (their Tab.1). However, after inspecting both the original papers and open-source repositories[12] of the aforementioned work we were unable to find any confirmation of the use of label smoothing.

## G ADDITIONAL SEGMENTATION RESULTS

In this section, we complete the picture by providing additional semantic segmentation results to Figs. 1 and 8. As in the main paper, the following segmentation maps are performed by ResNet-101-based DeepLab-v3+ trained on Cityscapes with cross-entropy and label-smoothing with $\alpha = 0.2$. We recall that the models and notebooks used to generate these plots are available on Hugging Face and GitHub, respectively. In these figures, we provide the rank of all the predictions – not only errors – to show the difference between the model's behavior on its errors but also correct predictions. The rank of the correct predictions ✓ is slightly greyed out compared to the rank of the errors ✗. Fig. 23 presents the full ranks corresponding to the scene used in the main paper.

Figs. 23 and 24 shows that the cross-entropy-based model predicts label maps in which pixel uncertainty ranks are generally smoother than those provided by the label-smoothing-based model. The label-smoothing model exhibits large areas with low uncertainty rankings, which suddenly increases near the boundary of objects. However, the boundaries are sometimes wrongly predicted, leading to (rank-wise) highly confident errors ✗. The pixels corresponding to the boundaries of objects, *where the probability of error is naturally higher* and which are most sensitive to (ground-truth) label errors, have higher confidence (lower uncertainty ranks) according to the LS-based model than the CE-based model. This explains the high-confidence errors ✗ in the misclassified boundaries of several objects, either in Figs. 1 and 24, and aligns with the imbalanced

---

[12]`https://github.com/google-research/vision_transformer`

suppression of Eq. (15). We again see how logit normalisation is able to decrease the confidence of the label-smoothing-trained model on highly confident (bright yellow) errors ✗.

We also provide Fig. 25 to show that the logit normalisation method is not always successful in solving the ranking of the predictions. In this figure, we see that a large part of the misclassified pixels (lower left) keeps low uncertainty ranks either using the CE-based model or the LS-based model while using logit normalisation. However, there are some improvements on the label-smoothing side, where the original low-ranked boundary errors ✗ are less (rankwise) confident, such as the right border of the median strip. Interestingly, LS-based high confidence errors in Fig. 25 mainly correspond to a change of texture of the median strip – likely associated with an uncertainty on the true label – and correlate with those of the CE-based model. The choices of $p$ match Fig. 13 from optimising AURC↓ on the 100 validation images.

## H  IMPACT AND FUTURE WORK

In this section, we provide some additional discussion about the (practical) impact of our work as well as promising directions for potential future research.

**Direct impact.**    By empirically verifying and analytically elucidating the limited experimental results pertaining to LS in (Zhu et al., 2024) (it is not the focus of that work), we provide strong evidence that the behaviour that LS degrades SC is *generalisable* (over architectures, data modalities *etc.*). In particular, we directly analyse the *loss*, which is common to all settings involving label smoothing. This will help inform practitioners of selective classification when they are designing and deploying systems. Furthermore, by explaining the efficacy of logit normalisation we provide an effective and well-motivated solution to the previously demonstrated problem. We emphasise that when logit normalisation was introduced in (Cattelan & Silva, 2024), it was not clearly explained why logit normalisation was effective on some pretrained models and ineffective on others. In our work, we analytically clarify the mechanism of logit normalisation, opening up the figurative black box, and are able to directly link it to our previous analysis on LS. This provides clear guidance on *when* and *why* to use logit-normalisation, giving potential practitioners *confidence* in the effectiveness of the approach, which is especially important in *high-risk safety-critical* applications.

**Future work.**    The empirical and analytical results relating to LS in our work naturally suggest that other training approaches that alter the labels such as Mixup (Zhang et al., 2018) may also have similar adverse effects and/or be amenable to similar gradient analysis. It also raises the question of how such label augmentations effect problem settings outside of selective classification such as OOD detection or transfer learning. The novel analysis of LS in Sec. 4 may also be useful in problem settings beyond SC: Does this aspect of label smoothing (suppressing the max logit less for incorrect predictions) affect generalisation? What about behaviour on OOD data? Could it help explain the behaviour in (Kornblith et al., 2021) where LS results in worse transfer/representation learning? Can this insight lead to a modification of LS to improve it? Our analysis of the mechanism of logit normalisation in Sec. 5.1 is also general – we simply demonstrate that logit normalisation reduces confidence when the max logit is higher. Thus, this knowledge can be potentially applied to other scenarios (*e.g.* OOD detection). Furthermore, in the case of post-hoc methods, logit normalisation using the $p$-norm is just one possible option for correcting the effect of LS – armed with the insight presented in our work it may be possible to develop superior or more principled methods for extracting better uncertainty estimates from logits.

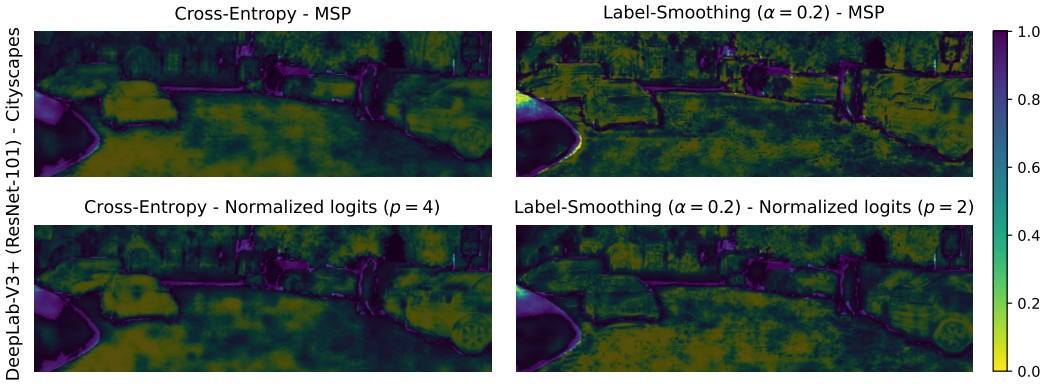

Figure 23: We provide all the predicted ranks on our segmentation plots to understand where the model predicted higher relative confidence. The pixel uncertainty rankings provided by the model trained with CE (left) are smoother than those of the model trained with LS $\alpha = 0.2$ (right).

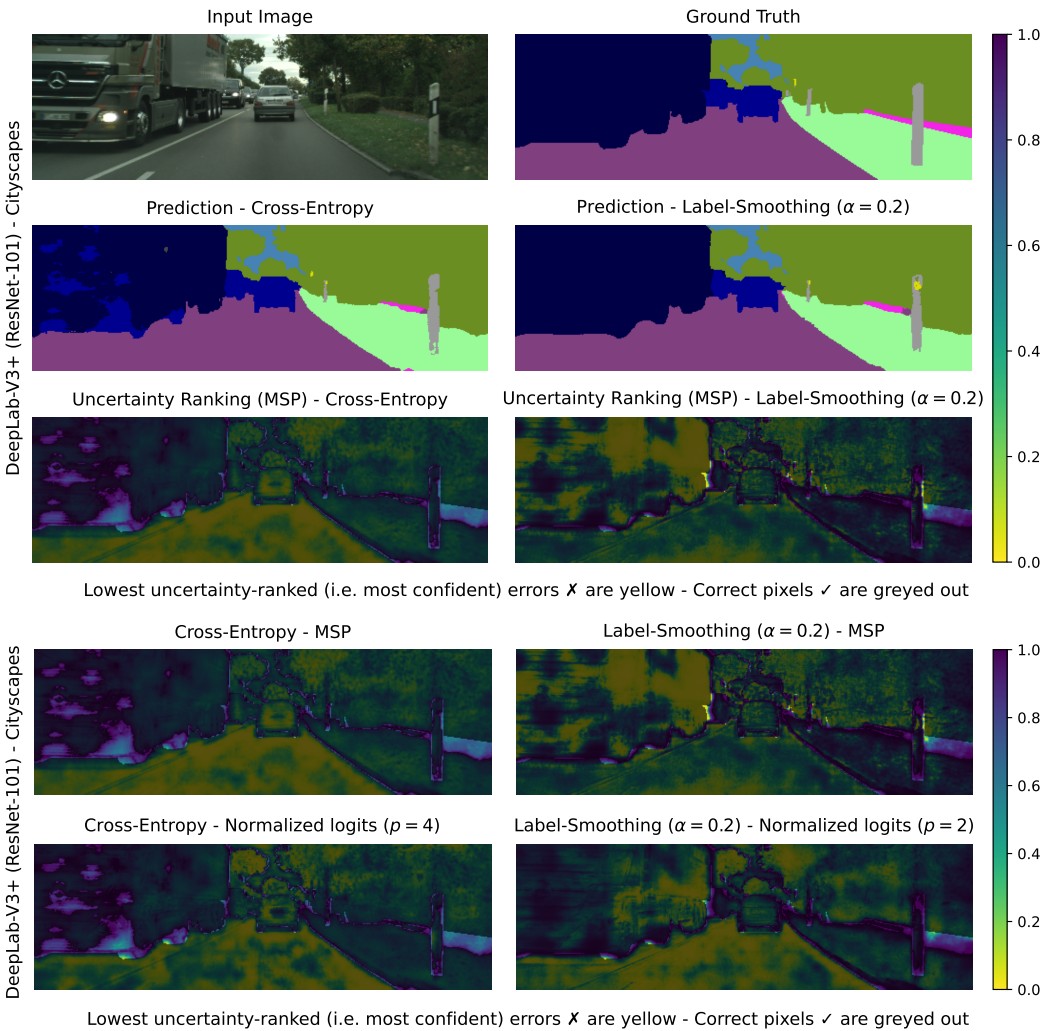

Figure 24: Logit normalisation improves on boundary-related very confident errors ✗, but does not completely fix the high (ranked) confidence of the bottom right errors ✗.

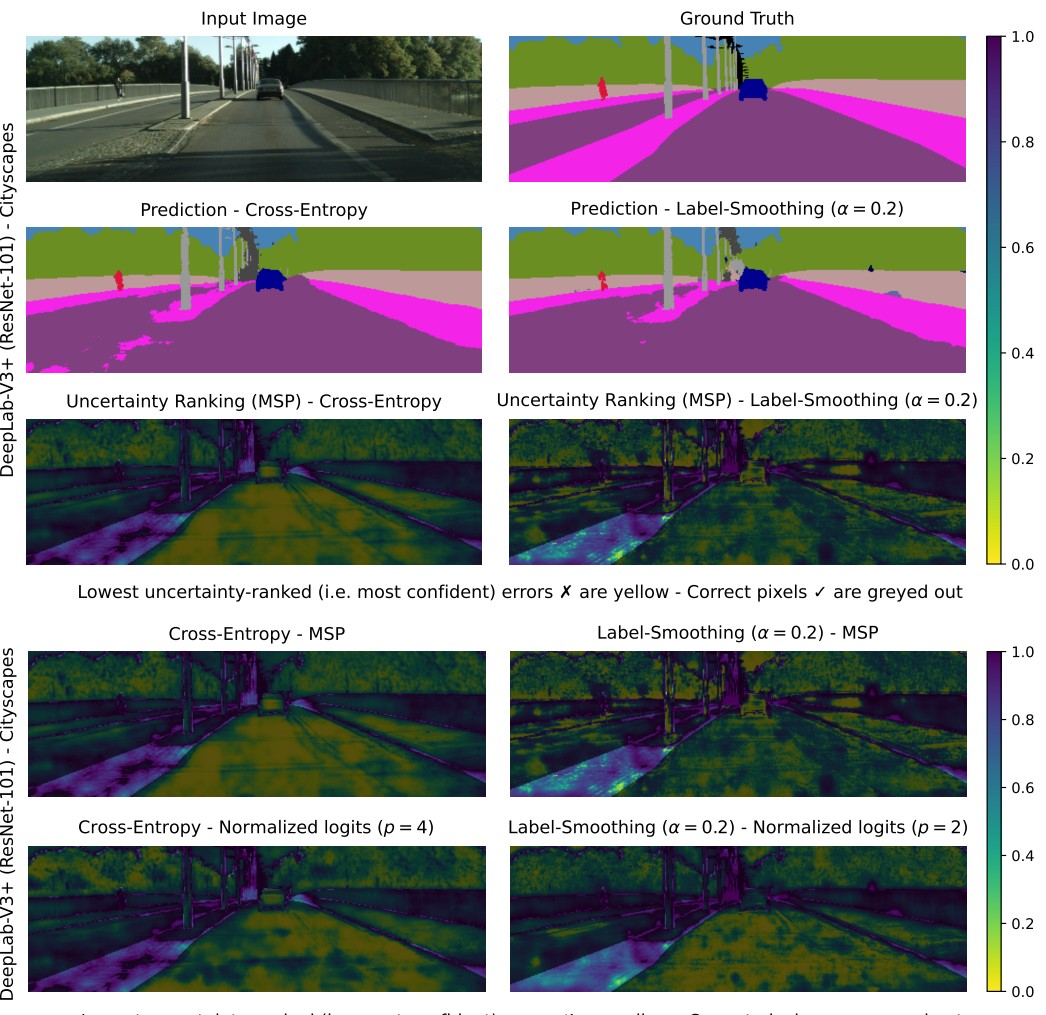

Figure 25: Logit normalisation does not always completely fix selective classification when used on a model trained with a label-smoothing (right).

