# OpenReview forum: "Towards Understanding Why Label Smoothing Degrades Selective Classification and How to Fix It"
_ICLR.cc/2025/Conference — ICLR 2025 Poster_

### Official Review · Reviewer_7wzD · 2024-11-01

**Soundness:** 3
**Presentation:** 2
**Contribution:** 3
**Rating:** 6
**Confidence:** 4

**Summary:**

Label smoothing (LS) is a popular regularization technique that improves test accuracy, yet previous research has shown that the LS models achieve degraded performance of selective classification (SC)
By comparing the formulation of Cross Entropy (CE) model and LS model, this work provides an explicit explanation on why this performance degradation is happening.
Building on the identified reasons for this degradation, the authors further discuss the impact of logit normalization and why it significantly improves SC performance of LS models but not that of CE models.
The primary contribution of this paper is the theoretical clarification of a previously unresolved issue. The experimental results are consistent with the theoretical findings presented in the paper.

**Strengths:**

Originality: the paper provides a very detailed and theoretic explanation for the unresolved issue of why label smoothing degrades selective classification.
Quality & Clarity: their explanations appear to be very reasonable. The theoretical gradient descent expressions align with the actual experimental results, making their work seem like a solid theoretical piece rather than some far-fetched guess.
Significance: The authors' explanation addresses a research gap and is likely to be valuable and insightful for future studies in selective classification.

**Weaknesses:**

While the authors present most of the logic and theory in an understandable and clear manner, unfortunately, the confusion in notation and concepts keeps me oscillating between being confused and understanding. The authors probably need to clarify some logic that they consider obvious but which I find uncertain. These points include:

1. Line 318: I would like to know more clearly the connection between the regularization in Equation 13 and the regularization gradient in Equation 14. Because, in my view, regularization refers to -\alpha/K, which is an item in L_LS, but the regularization gradient includes \alpha \bar{\pi}_k - \alpha/K, which is the result of L_LS - L_CE. The relationship between these two needs to be explicitly identified when they have similar names.
2. Line 319: There is a typo after the second equals sign in the formula.
3. Line 353: 'This directly impacts softmax-based U such as MSP', but I am not very clear on how. In fact, starting from Part 4.2, I completely lost track of the concepts of uncertainty, various expressions of pi, regularization, and P_error. This makes me feel like I understand but I don't.
4. Line 371: Again, due to the confusion on the concepts, 'v_max is more strongly suppressed for lower P_error' is not intuitive to me.
5. The caption for Figure 4 is confusing. Which one is the left, and which one is the right?
6. The caption for Figure 5 is confusing. What is the purpose of Figure 3? What does it illustrate? It is not clear.

**Questions:**

All my confusions have been listed under the weaknesses section. I welcome the authors to provide clarifications on points 1, 3, and 4.

---

> ### Author Response · Authors · 2024-11-16
> **Response to Reviewer 7wzD**
>
> We thank reviewer 7wzD for their feedback. We are especially thankful for the feedback on the clarity of the presentation in Sec. 4. Upon reflection, we agree that this should be improved. Addressing their specific queries:
>
> 1. We agree that the term “regularisation gradient” Eq. (14) is confusing. We originally conceived it in order to associate it with an intuition of “holding the model back” and also because LS can be understood as adding regularisation to CE. However, we understand that it may be confused with the “regularisation” in Eqs. (8,13). “Regularisation” in Eqs. (8,13) is illustrated in Fig. 2 (left), and refers to how label smoothing encourages the softmax output to be uniform. This is informal and mathematically distinct from the “regularisation gradient” in Eq. (14). As such **we have renamed “regularisation gradient” to “suppression gradient”**, to improve clarity as well as changed “regularisation” to “suppression” across relevant cases in the whole paper. This more directly reflects the idea of the gradient "pushing down" the logits.
> 2. We are unable to identify the typo, we would be grateful if the reviewer could point it out to us explicitly.
>
>
> 2.
>     - Although we define all notation in Sec. 2, upon reflection we agree that Sec 4 is difficult to parse given notation is not consistently clarified in the text. **We have included a glossary at the start of the appendix** such that a reader can more easily clarify any notation.
>
>     - *"This directly impacts softmax-based U such as MSP"*. If we consider the max softmax probability $\exp v_\text{max}/\sum_i \exp v_i$ in relation to logits $\boldsymbol v$, we can see that due to the exponentiation, MSP calculation will be dominated by the largest logits, in particular the max logit $v_\text{max}$.  Thus suppressing the max logit will tend to reduce the MSP. Other softmax-based uncertainties such as Entropy and DOCTOR (Appendix F.1 of the updated paper, prev E.1) are similarly dominated by the max logit and thus behave similarly. **We have added this clarification to Sec. 4.2**. We note that it may be helpful to intuit using "confidence" (negative uncertainty), i.e. pushing down on the max logit reduces model output confidence.
> 2. *“v_max is more strongly suppressed for lower P_error”*. In more plain language, the max logit is pushed down more on training samples where the model is (likely) right, and less pushed down when it is (likely) wrong. This leads the softmax to be less confident on correct predictions and more confident when it is wrong. Clarifying notation, Eq. (15) shows that for a given sample, compared to CE, LS suppresses/pushes down the maximum logit $v_\text{max}$ more(less) for lower(higher) probability of misclassification $P_\text{error}$.
> 2. Left refers to the **left two cells** and right the **right two cells**. We have separated them to make this clearer and updated the caption.
>
>
> 2.
>     - The purpose of Fig. 5 is to empirically illustrate/verify that training with LS leads to lower max logit values for correct predictions (following point 4.) We report max logit $v_\text{max}$ values *given* the MSP value $\pi_\text{max}$, because generally max logit (and MSP) for errors will be lower than correct predictions for both CE and LS. By conditioning on MSP we remove this bias. This also reflects that for the same $U$ (-MSP), samples with lower probability of error will have the max logit suppressed more by LS.
>     - Fig. 3 shows main selective classification results, which evidences that the ability of the MSP score to rank/separate/distinguish correct vs incorrect predictions is degraded when using LS compared to CE.
>
> We thank you again for your detailed feedback. Please do not hesitate to ask if you have any further queries, or require further clarification on the above.

---

> > ### Comment · Reviewer_7wzD · 2024-11-27
> > **Reply**
> >
> > In Equation 14, I think two $\delta$ are missing at $L_{LS} / \delta v_k - L_{CE} / \delta v_k$.
> >
> > I have no problem about other parts. The score is raised to 6. Good luck.

---

> > > ### Author Response · Authors · 2024-11-27
> > >
> > > Thank you for your service as a reviewer. We are grateful that you are satisfied with our response.
> > >
> > > The highlighted typo has been fixed in the updated submission pdf.

---

### Official Review · Reviewer_s8Gk · 2024-11-03

**Soundness:** 3
**Presentation:** 3
**Contribution:** 3
**Rating:** 8
**Confidence:** 4

**Summary:**

This paper explains why logit nomalisation improves SC performance for models trained with label smoothing.

**Strengths:**

1. I enjoy reading the paper. The presentation and organization look great.
2. This work provided both empirical and theoretic analysis to answer a research problem and considerably contributed to the field.

**Weaknesses:**

1. While the authors provided detailed experiments to demonstrate the properties of LS. I think it is also important to quantitatively compare the LS+logit normalisation with other existing solutions under the experimental setting used in this work to demonstrate the effectiveness of such a combination.

2. In figure 7(a), it is hard to find the "CE logit norm".

**Questions:**

N.A

---

> ### Author Response · Authors · 2024-11-16
> **Response to Reviewer s8Gk**
>
> We thank reviewer s8Gk for their feedback and heartwarming comments.
>
> > I think it is also important to quantitatively compare the LS+logit normalisation with other existing solutions under the experimental setting used
>
> If we consider post-training uncertainty scores for selective classification, we consider a number of other existing approaches (with and without LS) in Appendix F.1 (E.1 in previous version) . For softmax-based DOCTOR and Entropy, we find them to be similar but slightly worse than MSP. For OOD score Energy, we find it to be much worse than MSP. Thus LS+logit norm is better than LS+MSP and all the other uncertainty scores we investigate. Please let us know if we have interpreted your comment correctly.
>
>
> > In figure 7(a), it is hard to find the "CE logit norm".
>
> We have revised the figure for improved readability in the updated submission, by updating the colour of "CE logit norm".
>
>
> Please do not hesitate to ask if you have any further queries, or require further clarification on the above.

---

> > ### Comment · Reviewer_s8Gk · 2024-11-19
> > **Thanks for the reply.**
> >
> > My concerns are solved after reading the response, I will increase my score and confidence score.

---

> ### Author Response · Authors · 2024-11-19
>
> We thank you for your service. We are grateful that you enjoyed our paper and for the helpful feedback you provided.

---

### Official Review · Reviewer_KVT2 · 2024-11-04

**Soundness:** 2
**Presentation:** 3
**Contribution:** 3
**Rating:** 6
**Confidence:** 4

**Summary:**

This paper shows how label smoothing (LS) can negatively impact selective classification (SC) by combining empirical analysis on large-scale tasks with theoretical insights on logit-level gradients. The authors show that the degradation in SC performance worsens with increased LS strength, and they propose post-hoc logit normalization as an effective method for recovering SC performance lost due to LS.

**Strengths:**

- The paper provides a well-rounded analysis, both theoretical and experimental, demonstrating how LS degrades SC. By analyzing logit-level gradients, it addresses why LS has this adverse effect on SC, filling an important gap in understanding.
- Visualizations in the paper effectively illustrate the results, providing an intuitive understanding of how LS impacts SC performance.
- The findings have practical impact. By showing that LS leads to consistent degradation in SC, the paper suggests that it may partially explain the findings of strong classifiers surprisingly underperform on SC. They further shows that logit normalization can recover the degradation. This could be useful especially in the high-risk tasks like medical or robotics tasks.

**Weaknesses:**

The experimental analysis is limited to image classification and segmentation tasks, using only two specific datasets. This raises questions about the generalizability of the findings to other domains, such as text or tabular data, where label smoothing and selective classification may behave differently. Expanding the analysis to include diverse data types would strengthen the claim that label smoothing consistently degrades SC performance across various domains.

**Questions:**

- The experimental analysis is primarily focused on image classification and segmentation tasks, using two specific datasets. Could these findings generalize to other domains, such as text or tabular data?
- In Figure 3, the degradation impact of LS on SC decreases at high coverage. Is it possible to identify a threshold at which the effect of LS on SC begins to diminish?

---

> ### Author Response · Authors · 2024-11-16
> **Response to Reviewer KVT2**
>
> We thank reviewer KVT2 for their feedback and positive comments on our work.
>
> > using two specific datasets
>
> We note that Appendix B.3 contains results on CIFAR-100 and the jupyter notebook in the supplementary uses CIFAR-10. These results mirror the main paper's findings.
>
> > Could these findings generalize to other domains, such as text or tabular data?
>
> We have added binary classification results on two small-scale tabular UCI binary classification datasets (used by [1], for instance) in section B.4 of the appendix of the updated submission. We find LS degrades SC in this case as well.
>
> > Is it possible to identify a threshold at which the effect of LS on SC begins to diminish?
>
> Our interpretation of the reviewer’s query is that it relates to the SC rejection threshold $\tau$/corresponding level of coverage. The rightmost column of Fig. 3 shows the *difference* in risk between the baseline CE and LS. As you say, it shows that the *absolute* degradation in risk tends to diminish as coverage increases.
>  After a certain coverage ($\gtrsim 75\%$) the LS models may have lower risk than CE. This is due to the regularisation of LS improving the base accuracy (@100 coverage) of the model.
>
> The right of Fig. 3 shows that as coverage decreases LS consistently degrades risk *relative* to CE. This demonstrates that *LS is worse at separating errors from correct preds*, even if it has fewer errors @100 coverage
>
> We emphasise that LS prevents SC from being effective at low risks (Fig. 1 bottom), which need lower coverages to be achieved, which is especially important for safety-critical applications.
>
> We thank the reviewer again and would be grateful if the reviewer could clarify in the case we have misunderstood the above.
>
> Please do not hesitate to ask if you have any further queries, or require further clarification on the above.
>
> References:
>
> [1] Huang, X., Khetan, A., Cvitkovic, M., & Karnin, Z. (2020). Tabtransformer: Tabular data modeling using contextual embeddings. arXiv preprint arXiv:2012.06678.

---

> > ### Author Response · Authors · 2024-11-19
> > **Additional LSTM results**
> >
> > We have further added LSTM results on the IMDB [1] dataset in Appendix B.5. We believe that together with the MLP experiments on tabular data, this experimentally demonstrates that the negative effect of LS on SC generalises over data modalities and model architectures. We remark that this aligns with our mathematical analysis that focuses solely on the training loss (which is universal regardless of data modality or model architecture).
> >
> > We look forward to hearing back from you.
> >
> > [1] Maas et al. Learning Word Vectors for Sentiment Analysis, ACL 2011

---

> ### Comment · Reviewer_KVT2 · 2024-11-24
>
> I thank the authors for their detailed responses. Most of my questions have been answered. I will keep a positive score and raise my confidence to 4.

---

> > ### Author Response · Authors · 2024-11-27
> >
> > Thank you for your service as a reviewer. We are grateful that you are satisfied with our response.

---

### Official Review · Reviewer_wn1r · 2024-11-06

**Soundness:** 3
**Presentation:** 3
**Contribution:** 2
**Rating:** 6
**Confidence:** 3

**Summary:**

This paper investigates the impact of label smoothing (LS) on selective classification (SC), showing that while LS is a popular regularization technique for improving classification accuracy, it degrades SC performance. The authors empirically confirm this degradation across various models and tasks, then analyze the cause at a gradient level, finding that LS suppresses the highest logit differently based on prediction correctness. They propose post-hoc logit normalization as a solution, showing it effectively recovers SC performance degraded by LS.

**Strengths:**

**Originality**

The paper addresses a unique gap by investigating LS's unintended effect on SC. The logit-level gradient analysis provides a fresh perspective on why LS interferes with SC, helping bridge theoretical understanding with observed results.

**Quality**

The experiments are well-designed, involving diverse datasets (e.g., ImageNet, Cityscapes) and model architectures (e.g., ResNet-50, ViT) to thoroughly validate the findings. The analysis at both empirical and gradient levels strengthens the rigor, making the results compelling.

**Clarity**

The paper is well-organized, with clear visuals that illustrate LS’s effect on SC. Figures showing SC degradation and the effects of logit normalization make complex points more accessible.

**Weaknesses:**

My only concern is about the novelty of the contributions.

**Novelty concern**

* As admitted by the authors (Line 212), some of the core conclusions in the main paper are based on previous empirical observations (```for a single value of alpha LS degrades SC for CNN-based image classification```). And the introduced ```broader investigation``` draws the same conclusion as the literature.

* As specified by the authors (Line 416), another main contribution of the paper (```Logit Nomarlisation Improves the SC of LS-Trained Models```) also follows from the literature that ```logit normalization can improve the SC performance of many (but not all) pretrained models```.

**Questions:**

**Q1:** As observed by the authors, LS consistently leads to degraded SC performance, even if it may improve accuracy. What do authors think about the connection between NLS and SC, where NLS refers to Negative Label Smoothing introduced in R1.

**Q2:** In Line 132, it would be beneficial to include the definition of Kronecker delta.

**References:**

R1: To smooth or not? when label smoothing meets noisy labels. ICML 2022.

---

> ### Author Response · Authors · 2024-11-16
> **Response to Reviewer wn1r**
>
> We thank reviewer wn1r for their feedback.
>
> **Novelty/Significance**
>
> As noted by the reviewer, we are honest about the position of our contribution in the literature. We firmly believe that the additional *new knowledge* provided by our work will be of value to the research community of ICLR (and practitioners in general).
>
> *Label smoothing degrades selective classification*
>
> Although Zhu et al. (2022) empirically discover this, their empirical results are limited (it is not the focus of their work) and *there is no way of knowing whether these empirical results would generalise* or whether they are only true in specific instances.
> By providing extended empirical evidence, as well as a clear analytical explanation rooted in the mathematical loss, *we provide strong evidence that this behaviour is generalisable*. We believe that this insight is useful to researchers and practitioners of selective classification.
>
> *Logit normalisation improves the SC of LS-trained models*
>
> We provide an effective and *well-motivated* solution to the above problem, which is useful to practitioners.
>
> In the original logit normalisation paper (Cattelan & Silva, 2024)  they do not provide a clear explanation for *why* the approach is effective sometimes and isn’t effective in other instances. They suggest simply trying it out on a validation dataset and falling back to the MSP if it is not effective. The approach is presented like a black box. This may reduce the *confidence* of practitioners interested in using it, especially in *safety-critical applications* for which selective classification is relevant.
>
> By analytically investigating the mechanism of logit-normalisation, and linking it directly to our previous analysis/experiments on LS we are able to elucidate this black box. This provides clear guidance on *when* and *why* to use logit normalisation. This will give practitioners confidence in the effectiveness of the approach, when they previously may have chosen to avoid it in a safety-critical application.
>
> *Additional benefits of our analysis*
>
> The novel analysis in Sec. 4 raises questions for future work beyond selective classification, thus we believe it is of interest to the broader ICLR community: Does this aspect of label smoothing (suppressing the max logit less for incorrect predictions) affect generalisation? What about behaviour on OOD data? Could it help explain the behaviour in  (Kornblith et al., 2021) where LS results in worse transfer/representation learning? Can this insight lead to a modification of LS to improve it?
>
> Our analysis of logit-normalisation also generalises beyond LS – we simply demonstrate that logit normalisation reduces confidence when the max logit is higher. Thus, this knowledge can be potentially applied to other uncertainty scenarios (e.g. OOD detection).
>
>
> As authors, we believe such research, that aims to *explain* and *understand* behaviour in deep learning is valuable at ICLR.
>
>
>
>
> **Questions**
>
> 1. We investigated NLS and found training with it to be unstable. We remark that it appears worth investigating since negative $\alpha$ does reverse the logit suppression of Eq. (15). However, an analysis of the gradients (similar to the main paper) reveals that since NLS training targets can exist outside of the interval [0,1], in these cases the NLS logit gradients can never become zero (training minimum), leading to training instability. The discussion can be found in Appendix F.6 of the updated submission. We remark that NLS may still be useful for learning with label noise as presented in the original paper.
> 1. We have included the definition of the Kronecker delta in the paper (as well as the newly added notation glossary in the appendix)
>
>
> Please do not hesitate to ask if you have any further queries, or require further clarification on the above.

---

> > ### Comment · Reviewer_wn1r · 2024-11-26
> > **Thanks for your rebuttal**
> >
> > Thanks authors for the detailed discussion about the analysis of negative label smoothing.
> > In the revision, it would be better if the authors could include the above discussion of the novelty in the appendix as well.
> > The rating score is raised from 5 to 6. Good luck!

---

> > > ### Author Response · Authors · 2024-11-27
> > >
> > > Thank you for your service and further feedback.
> > >
> > > We have added the above discussion to Appendix H of the revised submission as requested.

---

### Author Response · Authors · 2024-11-16
**General Response to Reviewers**

Dear AC and reviewers, we thank you for your constructive comments and questions concerning our work. We are grateful that reviewers have expressed that

 - Our “experiments are well-designed” (**wn1r**), and “analysis at both empirical and gradient levels strengthens the rigor, making the results compelling” (**wn1r**).
 - Our paper has “clear visuals” that “make complex points more accessible” (**wn1r**) and help with “intuitive understanding” (**KVT2**).
 - Our findings fill “an important gap in understanding” (**KVT2**), “have practical impact” (**KVT2**), “considerably contributed to the field” (**s8Gk**) and are “likely to be valuable and insightful for future studies” (**7wzD**)

We have **updated the submission pdf** according to the reviews. **Changes are highlighted in blue-green** and are listed below.

 - **We have changed some references to the term “regularisation” to “suppression”**, in particular the “regularisation gradient” is changed to the “suppression gradient”. This is to remove a potential ambiguity and to improve clarity.
 - **We have added a glossary of notation at the start of the appendix**. This is to improve the readability of the paper, especially Sec 4.
 - We have also clarified the definition of Kronecker delta in the main paper and the glossary.
 - We have additionally re-written some of Sec. 4.2 to improve clarity (visible in blue-green). Fig. 7 has updated colours for improved visibility of CE logit norm.
 - **We include additional results on small scale tabular data** in Appendix B.4
 - **We include a discussion on Negative Label Smoothing [1]** in Appendix F.6

We thank the reviewers again for their detailed and thorough feedback. We welcome any further questions and look forward to addressing them swiftly.

*References:*
[1] To smooth or not? when label smoothing meets noisy labels. *In* ICML 2022.

---

> ### Author Response · Authors · 2024-11-19
> **Update**
>
> Dear reviewers, we have **further updated the submission pdf** with **LSTM experiments on text data** in Appendix B.5. They again show that LS degrades SC performance across different modalities and architectures.
>
> For reviewers who have yet to reply to our comments, we acknowledge the challenges of responding during this short time period. However, we are still eager to hear from you so that we can improve our manuscript and address any further issues/queries you may have.

---

### Meta-Review · Area_Chair_YzZj · 2024-12-16

**Metareview:**

The paper studies the effect of label smoothing (LS) on selective classification (SC) performance. Empirically, it is shown that LS systematically degrades the performance of the maximum softmax probability SC baseline. Analytically, it is shown that the reason for this can be traced to the gradient updates under LS favouring a stronger pull for samples with a lower inherent noise rate. This analysis is then extended to elucidate why logit normalisation can fare significantly better.

Reviewers were unanimously supportive of the paper. The work was found to be well-presented, intuitive, and of broad interest to practitioners. From the AC's reading, we tend to agree with this assessment.

Some critiques raised were that the work is limited to a few image classification datasets, and is as such more concerned on an analysis of known techniques (rather than proposing a new technique). For the latter, we tend to agree with the authors that such works are appropriate for ICLR, and should be of interest to the community. For the former, we agree that additional results for the empirical section (which is intended to be comprehensive) would be useful. The authors have added some results for tabular datasets which are a step towards this.

Overall, we believe this work is of interest to the community, and recommend its publication.

_Minor remark_: Appendix F.6 has some interesting analysis of "negative label smoothing". This appears related to the "backward correction" technique discussed in Lukasik et al., "Does label smoothing mitigate label noise?", ICML 2020.

**Additional Comments On Reviewer Discussion:**

Initial reviews were generally positive. Following the response, which provided additional results and clarifications, reviewers were unanimous in recommending acceptance.

---

### Decision · Program_Chairs · 2025-01-22

Accept (Poster)